# Mechanical coupling in the nitrogenase complex

Qi Huang[1], Monika Tokmina-Lukaszewska[2], Lewis E. Johnson[1,3], Hayden Kallas[4], Bojana Ginovska[1], John W. Peters[5], Lance C. Seefeldt[1,4], Brian Bothner[2], Simone Raugei[1]*

**1** Physical and Computational Sciences Directorate, Pacific Northwestern National Laboratory, Richland, Washington United States of America, **2** Department of Chemistry and Biochemistry, Montana State University, Bozeman, Montana United States of America, **3** Department of Chemistry, University of Washington, Seattle, Washington United States of America, **4** Department of Chemistry and Biochemistry, Utah State University, Logan, Utah United States of America, **5** Institute of Biological Chemistry, Washington State University, Pullman, Washington United States of America

* simone.raugei@pnnl.gov

**Data Availability Statement:** All relevant data are within the manuscript and its Supporting Information files.

**Funding:** Q.H., L.E.J., H.K., B.G., L.C.S. and S.R. were supported by the U.S. Department of Energy

## Abstract

The enzyme nitrogenase reduces dinitrogen to ammonia utilizing electrons, protons, and energy obtained from the hydrolysis of ATP. Mo-dependent nitrogenase is a symmetric dimer, with each half comprising an ATP-dependent reductase, termed the Fe Protein, and a catalytic protein, known as the MoFe protein, which hosts the electron transfer P-cluster and the active-site metal cofactor (FeMo-co). A series of synchronized events for the electron transfer have been characterized experimentally, in which electron delivery is coupled to nucleotide hydrolysis and regulated by an intricate allosteric network. We report a graph theory analysis of the mechanical coupling in the nitrogenase complex as a key step to understanding the dynamics of allosteric regulation of nitrogen reduction. This analysis shows that regions near the active sites undergo large-scale, large-amplitude correlated motions that enable communications within each half and between the two halves of the complex. Computational predictions of mechanically regions were validated against an analysis of the solution phase dynamics of the nitrogenase complex via hydrogen-deuterium exchange. These regions include the P-loops and the switch regions in the Fe proteins, the loop containing the residue β-188$^{Ser}$ adjacent to the P-cluster in the MoFe protein, and the residues near the protein-protein interface. In particular, it is found that: (*i*) within each Fe protein, the switch regions I and II are coupled to the [4Fe-4S] cluster; (*ii*) within each half of the complex, the switch regions I and II are coupled to the loop containing β-188$^{Ser}$; (*iii*) between the two halves of the complex, the regions near the nucleotide binding pockets of the two Fe proteins (in particular the P-loops, located over 130 Å apart) are also mechanically coupled. Notably, we found that residues next to the P-cluster (in particular the loop containing β-188$^{Ser}$) are important for communication between the two halves.

(DOE), Office of Science, Office of Basic Energy Science, Division of Chemical Sciences, Geosciences, and Biosciences. M.T.L., B.B. and J. W.P. were supported by the Biological and Electron Transfer and Catalysis (BETCy) EFRC, an Energy Frontier Research Center funded by the U.S. DOE, Office of Science (DE-SC0012518). Funding for the Proteomics, Metabolomics and Mass Spectrometry Facility used in this publication was made possible in part by the MJ Murdock Charitable Trust, MSU office of the VPREDGE, and the National Institute of General Medical Sciences of the National Institutes of Health under Award Number P20GM103474. The funders had no role in study design, data collection and analysis, decision to publish, or preparation of the manuscript.

**Competing interests:** The authors have declared that no competing interests exist.

## Author summary

Enzymatic function is often the result of a sequence of events, including precisely synchronized electron, proton and substrate delivery, which goes beyond the mere chemistry at the active site. To achieve this level of sophistication, enzymes must exploit a network of long-range communication through which chemical events, such as the binding of an effector to an allosteric site, hydrolysis of ATP, or electrochemical activity at a given location, affect on the function of a remote site. The nitrogenase complex vividly exemplifies the hierarchical complexity of enzymes. Nitrogenase catalyzes the reduction of atmospheric dinitrogen to ammonia, whose activity is regulated by long-range mechanical coupling between distant regions of the complex. After decades since the first structure of the nitrogenase complex was solved, this coupling network is still unclear. We used a computationally inexpensive, yet accurate approach, which is based on graph theory, to elucidate how distant regions of the protein complex communicate with each other. In particular, our study provides valuable insights on how the two halves of nitrogenase communicate and for this communication, what the most important regions are, laying out a strategy for future mechanistic studies.

## Introduction

Nitrogen is one of the most essential elements for life.[1] Although it is the most abundant element in the Earth's atmosphere, the overwhelming majority of atmospheric nitrogen is in the form of chemically inert dinitrogen ($N_2$) and unusable by most organisms.[2] In order to become biologically accessible, $N_2$ must be "fixed" to a more reactive compound, such as ammonia ($NH_3$). Nitrogen fixation in nature is accomplished by diazotrophic bacteria, such as those of genus *Azotobacter*, which contain the nitrogenase enzyme that catalyzes the reduction of $N_2$ to $NH_3$.[3, 4] Experimental evidence collected over the years indicates that nitrogenase activity is finely regulated by an intricate allosteric communication network that spans distances of nearly 100 Å.[5] In this work, we report an extensive analysis of the mechanical coupling in the nitrogenase complex using a graph-based methodology in which structural and dynamic signaling is used to establish mechanical connectivity and correlation. We identified how long-timescale, large-amplitude protein motions facilitate communication between distant protein regions.

Biological nitrogen fixation by nitrogenase follows the limiting stoichiometry in Eq 1,

$$N_2 + 8e^- + 16ATP + 8H^+ \rightarrow 2NH_3 + H_2 + 16ADP + 16P_i \tag{1}$$

with the obligatory formation of one $H_2$ molecule for every $N_2$ molecule reduced. Among the three known types of nitrogenases (molybdenum (Mo)-dependent, vanadium-dependent, and iron-only nitrogenase),[6–8] the Mo-dependent nitrogenase is most abundant and widely studied. The overall architecture and mechanism of the three nitrogenases are similar.[9–15] In this work, "nitrogenase" refers to Mo-dependent nitrogenase. The nitrogenase complex contains a catalytic molybdenum-iron (MoFe) protein and two electron-carrier iron (Fe) proteins (Fig 1). The MoFe protein is an $\alpha_2\beta_2$-heterotetramer, consisting of two catalytic $\alpha\beta$ units, each containing two iron/sulfur clusters: the [8Fe-7S] cluster (P-cluster) and the [7Fe-Mo-9S-homocitrate] catalytic cofactor (FeMo-co). The Fe protein is a $\gamma_2$-homodimer, containing a [4Fe-4S] cluster and two nucleotide-binding sites. Here, we will refer to the two monomers of the Fe protein as $Fe_A$ and $Fe_B$. Through a series of synchronized events, known as the "Fe protein cycle", each Fe protein delivers one electron at a time to the MoFe protein.[6] The Fe

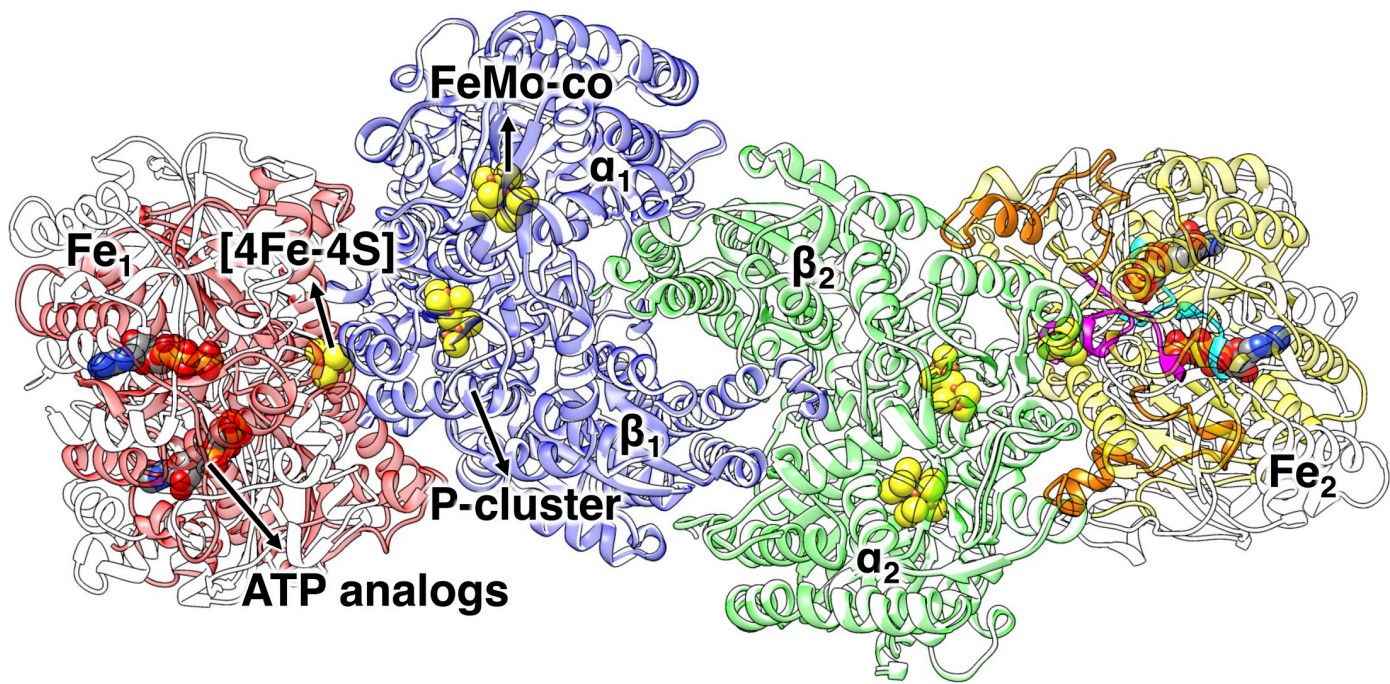

**Fig 1. Ribbon representation of the crystal structure of the ATP-bound nitrogenase complex (PDB ID: 2AFK).** $Fe_1$, red; $Fe_2$, yellow; $\alpha_1\beta_1$ unit of MoFe, blue; $\alpha_2\beta_2$ unit of MoFe, green. The ADP-bound complex (PDB ID: 2AFI [I]) shown in white, is aligned to the ATP-bound complex on the MoFe protein. In the ATP-bound complex, the P-loop (cyan), switch I region (orange) and switch II region (magenta) are highlighted in $Fe_2$, and the ATP analogues (AMPPCP), [4Fe-4S] clusters, P-clusters and FeMo-co are shown in spheres.

protein cycle needs to be repeated at least eight times to reduce one $N_2$ molecule to two ammonia molecules. The association of one Fe protein with two bound adenosine triphosphate (ATP) molecules to each $\alpha\beta$ half of the MoFe protein activates the electron transfer process. As elucidated by stopped-flow spectrophotometry, the first step is an intramolecular electron transfer event, from the P-cluster to the FeMo-co in the MoFe protein, and the second step is an intermolecular electron transfer, from the [4Fe-4S] cluster of the Fe protein to the oxidized P-cluster of the MoFe protein; the combination of these two steps is referred to as the "deficit-spending" model.[16] The two ATP molecules bound to the oxidized Fe protein are then hydrolyzed to two adenosine diphosphate (ADP) molecules,[17,18] with the release of two inorganic phosphates ($P_i$).[19] The dissociation of the Fe protein from the MoFe protein completes the Fe protein cycle.

There is growing evidence that delivery of electrons to the active site is strictly controlled by protein dynamics. A number of kinetic measurements have highlighted a sophisticated allosteric regulation of the electron transfer throughout the Fe protein cycle. The allosteric process includes intra-subunit communication, whereby binding of the Fe protein to one of the two $\alpha\beta$ units triggers an electron transfer within that $\alpha\beta$ unit,[20–22] and inter-subunit communication in the MoFe protein, whereby electron transfer in one subunit suppresses electron transfer in the other subunit.[5] Within each subunit of the nitrogenase complex, the first electron transfer event appears to be either gated by or associated with a conformational change upon docking of the Fe protein on the MoFe protein.[20–22] However, how this conformational change "gates" the electron transfer remains unclear. Measurement of the electron transfer rate as a function of the osmotic pressure suggested that transient large-scale protein conformational changes initiated after the Fe protein binding result in a transient increase of the solvent accessible surface of the nitrogenase complex of at least 800 Å$^2$. [20] It has been

proposed that this increase is associated with the transition of the complex from the pre-activated state (encounter state) to the ADP-bound state, rather than to the previously proposed activated state, i.e. the ATP-bound state.[23] The Fe protein is sensitive to the nucleotides binding and the docking of the ATP-bound Fe protein to the MoFe protein is likely to trigger the deficit-spending mechanism.[24] However, the P-cluster is ~16 Å from the Fe protein docking interface and the mechanism for signal transduction to gate the first electron transfer is unknown.[25,26] Many mechanistic studies have been performed to date to elucidate how the Fe protein docking gates the electron transfer. It has been proposed that ATP binding to the Fe protein provides the energy driving the electron transfer and stabilizes the protein-protein complex by excluding the previously-bound water molecules near the [4Fe-4S] cluster and the interface, respectively, and that the ATP-bound Fe protein has higher binding affinity to the MoFe protein than the ADP-bound one.[27] ATP binding to the Fe protein also moves the [4Fe-4S] cluster a few ångströms closer to the protein surface,[28,29] which may change the driving force or donor-acceptor coupling for the electron transfer.[30] An alternative model suggests that the energy associated with the Fe protein/MoFe protein docking is transduced to electrochemical energy through changes in the ligation state of the P-cluster or FeMo-co.[31]

Comparison of the crystal structures of the unreactive ATP-analog-, adenosine 5'-(β,γ-methylene)triphosphate (AMPPCP), and ADP-bound nitrogenase complexes reveals a rigid body reorientation of the Fe protein on the surface of the MoFe protein upon ATP hydrolysis. [24,28] Inspection of the crystal structures indicates that: (i) the binding and hydrolysis of the nucleotides in the Fe protein influences the Fe protein/MoFe protein interaction through residues 39–69 of each Fe protein monomer (Switch I region, Fig 1),[32] (ii) the nucleotide-binding sites (residues 9–16, referred to as the P-loop) are coupled with the [4Fe-4S] cluster through the Fe protein residues 125–135 (Switch II region, Fig 1), [25] and (iii) the [4Fe-4S] cluster moves relative to the Fe protein/MoFe protein docking interface.

In contrast, available crystal structures of the nitrogenase complex with either AMPPCP or ADP show no clear changes in the MoFe protein with respect to the isolated protein.[28,29,33] However, binding of the Fe proteins and hydrolysis of ATP can induce changes in the equilibrium dynamics of the MoFe protein favoring or initiating electron transfer, whereby perturbations within the Fe protein are propagated to residues near FeMo-co and/or P-cluster in the MoFe protein, which may not be evident from the mere inspection of the crystallographic structures. In support of this hypothesis, normal mode analysis using an anisotropic network model (ANM) has indicated that three residues in the MoFe protein (α-64$^{Tyr}$, β-98$^{Tyr}$, β-99$^{Phe}$), which play a role in electron transfer from the P-cluster to the FeMo-co, are coupled to the motions of the Fe protein.[34] In addition, it is shown both experimentally and computationally that the P-cluster changes its ligation state by binding the β-188$^{Ser}$ upon one-electron oxidation.[35] Recently, it has been proposed that the role of β-188$^{Ser}$ is to stabilize the oxidized P-cluster, based on the observation that β-Ser188Ala mutant shows similar activity as the wild-type.[36] Interestingly, β-Ser188Gly mutation appears to lose ~40% specific activity.[37] It is still unclear if β-188$^{Ser}$ or residues nearby are affected by the binding of the Fe protein or coupled with the ATP hydrolysis.

Pre-steady state kinetic data suggests that, although both halves of the nitrogenase are active, electron transfer in one half of the nitrogenase complex (*e.g.*, Fe$_1$-α$_1$β$_1$) inhibits electron transfer in the other half (*i.e.*, Fe$_2$-α$_2$β$_2$).[5] An ANM covariance analysis of the displacement of the residues also indicated negative correlation in the motions of the two subunits. It is proposed that the dynamics mentioned above might gate the electron transfer in one subunit and inhibit that in the other subunit.

Toward an understanding of the allosteric regulation of electron transfer in nitrogenase, we have performed an extensive analysis of the mechanical coupling in the ATP- and ADP-bound

nitrogenase complex. The present study relies on a graph-based method for investigating the mechanical coupling between distant sites in proteins developed in our group.[38] Mechanically coupled regions of the nitrogenase complex as inferred computationally were validated against hydrogen/deuterium (H/D) exchange experiments.

For sites in a protein to be mechanically coupled (causality of the correlation), there must be correlated motions as well as a plausible path for the communication between the sites. Our method is able to identify the correlated and connected motions, as well as the importance of specific residues along the mechanical coupling pathways. The method has been shown to provide semiquantitative information on the regulatory importance of specific structural elements in the mechanical coupling between distant parts in the allosterically regulated human liver pyruvate kinase.[38] Here, we extend its application to the nitrogenase complex in order to study the communications within and between the subunits of the complex. We show that the Fe protein and the MoFe protein are indeed mechanically coupled and that regions of high coupling involve structural elements proposed to be key for allosteric communication in the nitrogenase complex. Specifically, the P-loop in the Fe protein, residues adjacent to β-188$^{Ser}$ in the MoFe protein (in particular β-189$^{Phe}$), as well as residues at the Fe protein/MoFe protein interface, were found to be strongly coupled. In addition, the P-loop and the switch regions for the two Fe proteins in the nitrogenase are also mechanically coupled to each other, which suggests a mechanically mediated long-range communication between the two halves of the nitrogenase complex. Important regions in the MoFe protein coupled to the Fe protein identified by our method were validated against an analysis of the solution phase dynamics of the protein complex via hydrogen-deuterium exchange experiments on the free MoFe protein in solution and in complex with the ATP-bound Fe protein.

## Methods

### Mechanical coupling analysis

Mechanical coupling analyses were conducted on the crystal structures of the ADP-bound nitrogenase complex (PDB ID: 2AFI[28], 3074 residues solved) and the AMPPCP-bound nitrogenase complex (PDB ID: 2AFK[28], 3064 residues solved). While the 2AFK structure is currently replaced by 4WZB, the major difference between the 4WZB and 2AFK structures is the nature of the FeMo-co central atom (C vs. N). The position of the Cα atoms, which are used in our normal mode analysis, is very similar in the two structures (RMSD = 0.18 Å). We adopted the 2AFK structure for modeling the ATP-bound complex for consistency with the structure adopted for the ADP-bound complex (2AFI structure). In the following, when discussing the computational results based on the above crystal structures, we refer to the nonhydrolyzable ATP-analog structure as the "ATP-bound" nitrogenase. In the present study, the explicit presence of the cofactors was neglected. Inclusion of the cofactors as done in earlier studies yields negligible differences in the covariance matrix, and the analysis based on it does not change.[5] The number of residues in 2AFK and 2AFI PDB structures is slightly different (3064 vs. 3074). Differences are limited to the (unstructured) termini regions and are not significant for the present analysis. In order to fully utilize the crystallographic information, we included all the residues contained in the crystallographic structures. However, analyses were performed only for the common residues. The general protocol adopted for the mechanical coupling analysis is described briefly here; further details can be found in our previous work. [38]

Our analysis is based on a coarse-grained ANM[39] as implemented in ProDy.[40] Coarse-grained ANM calculations are definitely more amenable for the type of analysis presented here than a principal component analysis based on molecular dynamics trajectory. Indeed, the size

of the nitrogenase complex poses serious limitations on the convergence of long-time, large-amplitude motions, which are crucial for the long-range communication.

The normalized covariance matrix of interatomic motions $\mathbf{C} \equiv \{C_{ij}\}$ and the Euclidean distance matrix $\mathbf{D} \equiv \{D_{ij}\}$ are generated from the coordinates of the $C_\alpha$ atoms. The first 6 trivial modes (rotations and translations) are excluded. Using the covariance matrix, the adjacency matrix $\mathbf{A} \equiv \{A_{ij}\}$, describing the network of the correlated motions in the protein is introduced as

$$A_{ij} = 1 - |C_{ij}|. \tag{2}$$

In order to remove the trivial correlations, the matrix $\mathbf{A}$ is pruned according to the rule

$$A_{ij} = 0 \text{ if } C_{ij} < (\mu_{|C|} + \sigma_{|C|}) \text{ or } D_{ij} > D_{\text{cut}}, \tag{3}$$

where $\mu_{|C|}$ and $\sigma_{|C|}$ are the mean and standard deviation of $\mathbf{C}$, respectively, and $D_{\text{cut}}$ is the contact distance cutoff set to 11 Å, which is the minimum distance to generate a direct path between any two given residues. The maximum coupling threshold $C_{\text{thresh}}$ is set to 0.95, so any $|C_{ij}| > C_{\text{thresh}}$ is set to $C_{\text{thresh}}$, and $A_{\text{thresh}} = 1 - C_{\text{thresh}}$. Such a threshold is applied for numerical stability to avoid small elements in the $\mathbf{A}$-matrix; the value is based on expected coupling within secondary structural elements.[38] The $\mathbf{A}$ matrix corresponds to a two-dimensional undirected graph, in which each node represents one residue, and the length of each edge connecting two nodes represents the strength of coupling between two residues. From the matrix $\mathbf{A}$, the geodesic matrix $\mathbf{G}$ is generated as the sum of all edge lengths along the shortest path from node $i$ to node $j$, using MATLAB.[41] We indicate the shortest path in the covariance space as the most efficient pathway between $i$ and $j$. Because of the elongated shape of the nitrogenase complex, the shortest path in the covariance space is often similar (but not identical) to the shortest path in the cartesian space. Finally, the mechanical coupling matrix $\mathbf{M}$ is calculated from the correlation and geodesic matrices as

$$M_{ij} = A_{thresh} \frac{|C_{ij}|}{G_{ij}}. \tag{4}$$

In the proposed mechanical coupling analysis, we filter out the purely coincidental correlated or anticorrelated motions inferred from the covariance analysis by weighting each element $C_{ij}$ of the covariance matrix by the geodesic distance matrix in the covariance space between residue $i$ and residue $j$. By doing so, we focus on how the motions of the nitrogenase complex about one equilibrium conformation allow for the communication between the distant sites. In this spirit, "mechanical coupling" here indicates that two residues or two regions not only have correlated dynamics, but, more importantly, there is an efficient allosteric communication pathway connecting them (*i.e.*, short geodesic distance). However, it does not necessarily indicate any functional role, whose assessment is clearly beyond the coarse-grained description of the protein adopted here.

With the mechanical coupling graph of the protein at hand, the shortest path trees can be generated from a given site X to any other residue in the protein or between two given sites X and Y using MATLAB.[41] In the all-residue representation, the site of each [4Fe-4S] cluster includes four cysteines, the site of each P-cluster includes six cysteines, and the site of each FeMo-co includes one cysteine and one histidine.

## Communication networks

The mechanical coupling matrix $\mathbf{M}$ highlights parts of the nitrogenase complex that have correlated motions and are coupled by efficient pathways between them. These pathways are

dictated by the protein scaffold, which connect distant regions through delocalized vibrational modes. However, the matrix **M** does not directly provide any information on the identity of the residues that contribute to the mechanical coupling between two distant parts because this information is abstracted into the geodesic matrix **G**. Typically, there is a multitude of pathways connecting distant parts of a protein. We recently showed that these pathways may share a few common residues that are points of high traffic (many pathways passing through them) and represent effective "chokepoints" for the allosteric communication.[38] The contribution of each residue to the mechanical coupling can be obtained by calculating the shortest-path tree from a given region (*e.g.*, the P-loop in the Fe protein) to another region (or all other residues) and merging the resulting trees to form a graph of strong coupling. The importance of a given residue can be then quantified by the number of pathways passing through it *vs.* the total number of possible pathways as measured by the cost-weighted betweenness centrality $\chi^X$ for each residue or $\chi^{X\text{-}Y}$ for any pair of residues. The cost-weighted betweenness centrality quantifies the influence of a residue on the mechanical coupling pathways originating from (or ending to) a given site X or between two sites, X and Y, of the protein. Therefore, perturbation of a residue with larger cost-weighted betweenness centrality is expected to have larger impact on the coupling, or more control over the allosteric network, because much of the allosteric signaling will pass through that residue.

## Significance of individual normal modes for the strongest coupling pathways

The covariance matrix is given by the summation of all normal modes. In order to study the contribution of an individual normal mode to the strongest coupling pathways in the nitrogenase, a sub-covariance matrix $\mathbf{C}_{-k}$ can be obtained by subtracting the $k$-th mode from the full covariance matrix:

$$\mathbf{C}_{-k} = \mathbf{C} - \frac{1}{\lambda_k} \boldsymbol{v}_k \boldsymbol{v}_k^T, \tag{5}$$

where $\lambda_k$ and $\boldsymbol{v}_k$ are the $k$-th eigenvalue and eigenvector of the covariance matrix obtained from the ANM. Then $\mathbf{C}_{-k}$ is normalized by the same mean-square fluctuations of the full covariance matrix, and the same procedures and cutoffs as above are applied to construct coupling pathways without the respective mode. Removing a significant mode will lead to a large change on the coupling pathways. The influence of an individual mode $k$ can be quantified by the inner product of the betweenness of the nodes (*i.e.*, residues) in the coupling pathways constructed from the sub-covariance matrix $\mathbf{C}_{-k}$, *i.e.*, the vector $(\chi_{-k,1}, \chi_{-k,2}, \cdots, \chi_{-k,N})$, and the betweenness of the nodes in the strongest coupling pathway constructed from the full covariance matrix, *i.e.*, vector $(\chi_{0,1}, \chi_{0,2}, \cdots, \chi_{0,N})$. In order to take into account both the directionality and the magnitude of the two vectors, the overlap of the coupling pathways before and after removing mode $k$, $O_k$, is normalized as

$$O_k = \frac{\sum_{i=1}^{N} (\chi_{-k,i} \cdot \chi_{0,i})}{\sum_{i=1}^{N} (\chi_{0,i} \cdot \chi_{0,i})}, \tag{6}$$

where $N$ is the total number of nodes. In fact, this overlap analysis can be used to identify the similarity of coupling pathways with respect to $\chi$ vectors between any two motions/displacements. The influence of the mode $k$ is quantified as

$$I_k = 1 - O_k. \tag{7}$$

The larger the impact of normal mode $k$, the higher the value of $I_k$ is. The limiting value $I_k = 1$

implies that one single normal mode $k$ gives rise to all necessary motions for the coupling. $I_k$ can be smaller than 0 (*i.e.*, $O_k$ larger than 1), which indicates that removing the normal mode $k$ may enhance the importance of specific residues.

ProDy is used to generate the normal modes and the Normal Mode Wizard (NMWiz) plugin to VMD is used to visualize the normal modes.[40]

### Hydrogen/deuterium exchange experiments

Nitrogenase proteins were expressed and purified from *Azotobacter vinelandii* strains DJ995 (wild-type MoFe protein with His tag) and DJ884 (wild-type Fe protein) in anaerobic conditions as previously described.[42] Protein purity was assessed by SDS-PAGE analysis with Coomassie Blue stain used for protein detection. Protein concentration was determined by biuret assay against a bovine serum albumin standard.

The H/D exchange reaction was conducted in anaerobic conditions as described elsewhere. [43] Briefly, four solutions were prepared: deuterated buffer (50 mM Tris, 12 mM dithionite, pD 7.4), non-deuterated buffer (50 mM Tris, 12 mM dithionite, pH 7.4), quench solution (3% formic acid; FA, Sigma), and dilution buffer (15 mM ATP and 8 mM $MgCl_2$) MoFe protein in $H_2O$ was diluted 10-fold during mixing with Fe protein in a $D_2O$ dilution buffer. Samples were removed and quenched to stop exchange after 1 hour and 24 hours. Reactions were quenched by placing 10 μL of sample into a mixture of 3% formic acid and porcine pepsin (Sigma, 0.2 mg/mL final concentration) on ice. After a two-minute incubation with pepsin, the reaction sample was frozen in liquid nitrogen and stored at -80˚C until liquid chromatography-mass spectrometry (LC-MS) analysis.

A 1290 UPLC series chromatography system (Agilent Technologies) coupled to a 6538 UHD Accurate-Mass Q-TOF LC-MS mass spectrometer (Agilent Technologies). MassHunter Qualitative Analysis version 6.0 (Agilent Technologies) was used to process raw data. Peptide identification used MS and MS/MS analysis which was augmented with Peptide Analysis Worksheet (PAWs, ProteoMetrics, LLC.). Peptide exchange values and deuterium uptake curves were generated using HDExaminer (Sierra Analytics, Inc.). Percent deuterium (%D) incorporation at each time point was calculated by dividing the number of deuterium atoms incorporated (#D) by total deuterium incorporated at 24 hours. Overlapping peptides were resolved as described by Pascal et al.[44] The investigation of the correlations between protein dynamics based on the difference in percent of deuterium exchange between each region of the MoFe protein was done in R 4.0 language and environment for statistical computing and graphics (R Development Core Team, 2020) using corrplot package.[45]

The analysis of the HDX data was a two-part process. First, compare deuterium incorporation in all conditions, the deuterium uptake was determined by monitoring shifts of the centroid peptide isotopic distribution in the mass spectra. The percent deuterium (%D) incorporated at a given condition was calculated by dividing the number of deuterium atoms incorporated by the number of deuterium atoms incorporated after 24 hours. Our investigation of the protein dynamics correlations and correlation patterns was based on the difference in %D between each region of the MoFe protein. The analysis was carried out using the symmetric matrix $D_{ij} = |d_i - d_j|$, which is defined in terms of the absolute values of the difference in the deuterium content %D of each peptide pairs ($d_i$ and $d_j$). The correlation matrix was generated using Pearson correlation. To group patterns based on the correlation of difference in the matrix elements $D_{ij}$, hierarchical clustering was used (1-corr formula as distance and complete linkage as agglomeration method). The number of useful clusters (distinct patterns of protein dynamics correlations) was established based on k-means clustering.

## Results and discussion

### Covariance and mechanical coupling analysis

Analysis of available crystal structures does not show conformational changes in the MoFe protein upon association with either the ATP- or ADP-bound Fe protein. Crystal structures provide us with a picture of the equilibrium structure and, clearly, do not inform us on potential transient changes. The root mean-square-deviation (RMSD) between the ATP- and ADP-bound complex is 6.6 Å. RMSD between the MoFe proteins complexed with the ADP- or ATP-bound Fe protein is ~0.3 Å, while the RMSD of the ADP- and ATP-bound Fe protein itself is ~2.5 Å (S1 Table). This shows that the two complexes at their corresponding equilibrium conformation captured by crystallography differ in the Fe proteins position through a rigid body reorientation of the Fe protein on the surface of the MoFe protein.[24,28]

The covariance matrix of the displacements of the residues **C** for both the ATP- and ADP-bound nitrogenase complex is reported in S2 Fig. The covariance matrix shows a large degree of correlation between the motions of various regions of the nitrogenase complex. The motion of the two αβ units of the MoFe protein is mostly positively correlated, whereas with the Fe proteins they are mostly negatively correlated. However, interesting exceptions were found. For instance, the residues near the cysteinate ligands of the P-cluster are positively correlated with the residues near the [4Fe-4S] cysteinate ligands (residue 97, 132) of the Fe proteins. The two Fe proteins are largely positively correlated, but they show negative correlation, *e.g.*, near the [4Fe-4S] cysteinate ligands.

The mechanical coupling matrix **M** (Fig 2) is able to single out among the many regions with highly correlated or anti-correlated motions those that are also mechanically coupled through efficient paths. The magnitude of the mechanical coupling, as described by the matrix **M**, depends on the geodesic matrix. By construction, residues with a large covariance and geometrically close to each other have short geodesic distances, and thus, are strongly mechanically coupled. In addition, within each unit, the coupling is generally strong as a consequence of its compactness. Several correlated regions in the covariance matrix also appear in the mechanical coupling matrix (Fig 2), which indicates that the correlated motions between the

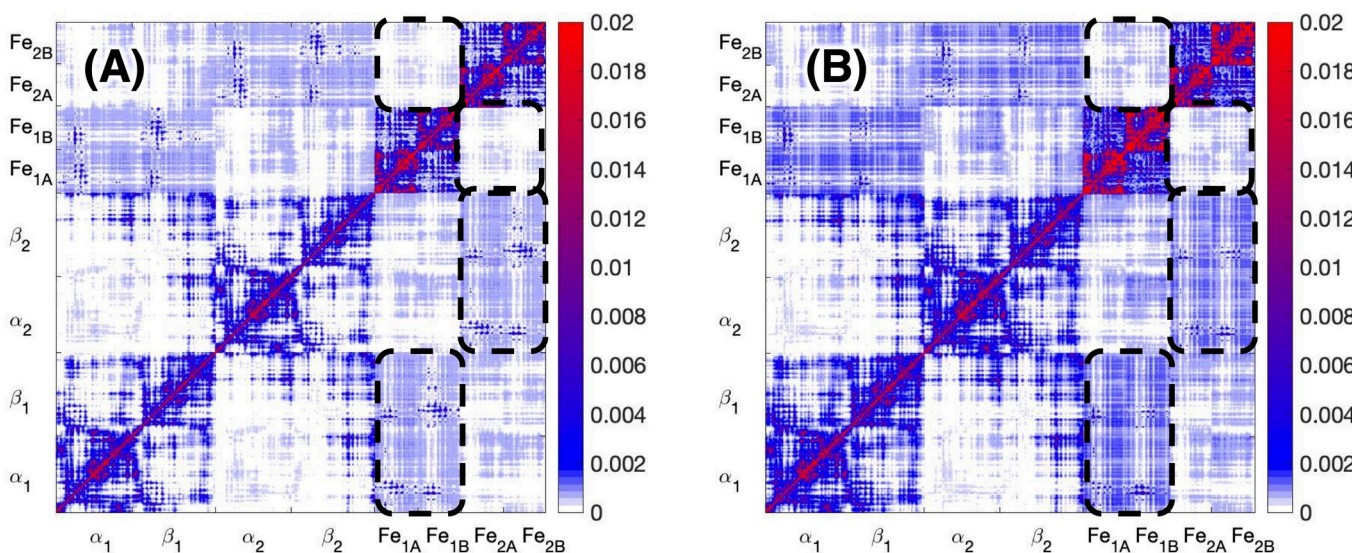

**Fig 2. Mechanical coupling matrix M for (A) the ATP-bound nitrogenase complex and (B) the ADP-bound nitrogenase complex**. The mechanical couplings between Fe₁ and α₁β₁, Fe₂ and α₂β₂, Fe₁ and Fe₂ are highlighted in dashed blocks.

two Fe proteins, between the two units of the MoFe protein, and between the Fe protein and the MoFe protein, have efficient mechanical routes connecting them. Overall, the ADP-bound complex has a slightly stronger mechanical coupling than the ATP-bound complex (S1 Fig, *i. e.*, the difference between Fig 3A and 3B), with the largest differences within the Fe protein or at contact points between the Fe and MoFe proteins. Due to the general similarity between the ADP- and ATP-bound complex, in the following, the mechanical coupling between the various units of the ATP-bound nitrogenase complex is mainly discussed. The exceptions will be explicitly discussed as appropriate.

Within each Fe protein, the P-loops are coupled to the protein-protein interface and the [4Fe-4S] cluster through the switch region I and II, respectively. Liao and Beratan discussed this coupling in their computational analysis of the isolated Fe protein.[46] Using a coarse-grained model, they identified correlated regions in the Fe protein and reported that residues in the P-loops and switch regions are relatively rigid in the slowest normal mode, which can be important for function-relevant conformational change. As we will discuss below, we show that there is indeed an efficient communication pathway between these regions, which, in the nitrogenase complex, extends up to the interior of the MoFe protein. The mechanical coupling matrix **M** between $Fe_1$ and $\alpha_1\beta_1$ for the ATP-bound complex is reported in Fig 3A. Three $\alpha$-helices in the $\alpha$ (helices $\alpha_7$, $\alpha_9$ and $\alpha_{10}$ following the numbering in the 2AFK PDB file) and the corresponding helices in the $\beta$ component (helices $\alpha_{33}$, $\alpha_{35}$ and $\alpha_{36}$) of the MoFe protein are strongly coupled to the $Fe_1$ protein. The $\alpha_1\beta_1$ region involved in the coupling extends from these $\alpha$-helices located at the docking interface near the P-cluster to the [4Fe-4S] cluster in the Fe protein. In particular, in the ATP-bound complex the loop containing $\beta$-188$^{Ser}$ (residues 186–190) is highly coupled with the switch regions of the Fe protein (Fig 3A). This coupling is negligible in the ADP-bound nitrogenase (S3A Fig).

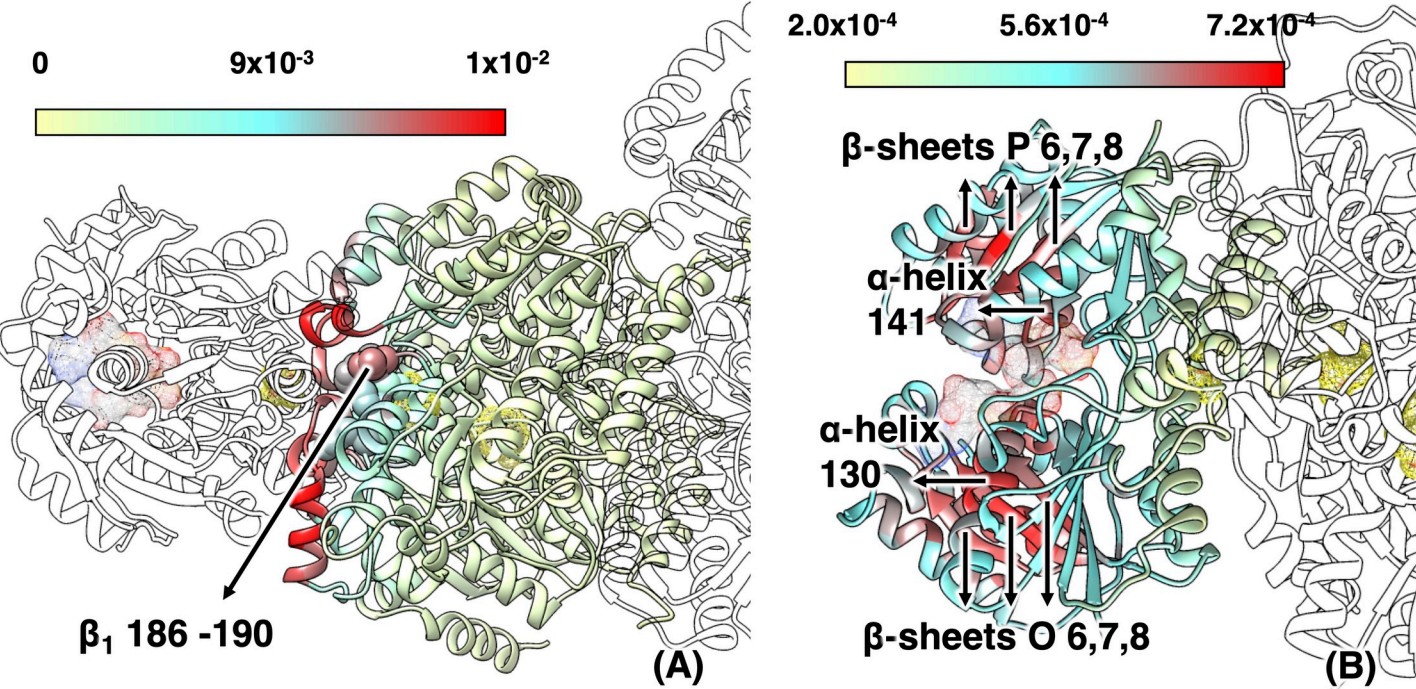

**Fig 3. Maximum mechanical coupling M of (A) residues in $\alpha_1\beta_1$ to any residue in $Fe_1$ and (B) residues in $Fe_2$ to any residue in $Fe_1$ in ATP-bound complex.** The ATP analogues, [4Fe-4S] clusters, P-clusters, and FeMo-co are shown in mesh surface. Residue 186–190 in $\beta_1$ are highlighted in spheres in (A); one $\alpha$-helix and three $\beta$-sheets in each monomer of $Fe_2$ appear red in (B) (following the numbering in the 2AFK PDB file).

The mechanical coupling between the two Fe proteins is shown in Fig 3B. The coupling is relatively weak by construction, because of the long coupling pathway, *i.e.*, a long geodesic distance in the covariance space. However, it highlights very specific coupling patterns. The residues in one Fe protein with the highest coupling to the residues of the other Fe protein are located in the P-loops, the α-helix (helix $\alpha_{130}$ in $Fe_{2A}$ and $\alpha_{141}$ in $Fe_{2B}$) and the β-strand (strand $O_5$ in $Fe_{1A}$ and strand $P_5$ in $Fe_{1B}$) that the P-loop is bridging, along with three other β-strands in the same β-sheet (strands $O_6$, $O_7$, $O_8$ in $Fe_{2A}$ and strands $P_6$, $P_7$, $P_8$ in $Fe_{2B}$). All these structural elements envelope the nucleotide binding sites. This result suggests that a perturbation introduced in the nucleotide binding pockets (for instance, hydrolysis of the nucleotide or loss of the inorganic phosphate $P_i$) is propagated to the nucleotide binding sites in the opposite unit. The mechanical coupling in the ADP-bound complex (S3B Fig) is similar to the ATP-bound complex, but it appears to be smaller in the ATP-bound complex (26% smaller in average). These differences are consistent with the covariance matrix, which indicates that the ADP-bound complex has a more flexible structure.

## Changes in protein flexibility and mechanical coupling upon ATP hydrolysis

As we highlighted above, the structure of the MoFe protein does not provide information for its change upon ATP hydrolysis. However, the rearrangement of the Fe protein on the MoFe protein upon hydrolysis influences the flexibility of various structural elements close to the Fe protein/MoFe protein interface. The influence of the Fe protein reorientation upon ATP hydrolysis was quantified in term of changes in the root mean-square fluctuation (RMSF) of the $C_\alpha$ atoms as obtained from the ATP- and ADP-bound structures. The RMSF analysis presented in Fig 4 shows that, apart from a trivial perturbation in the dynamics of the residues at

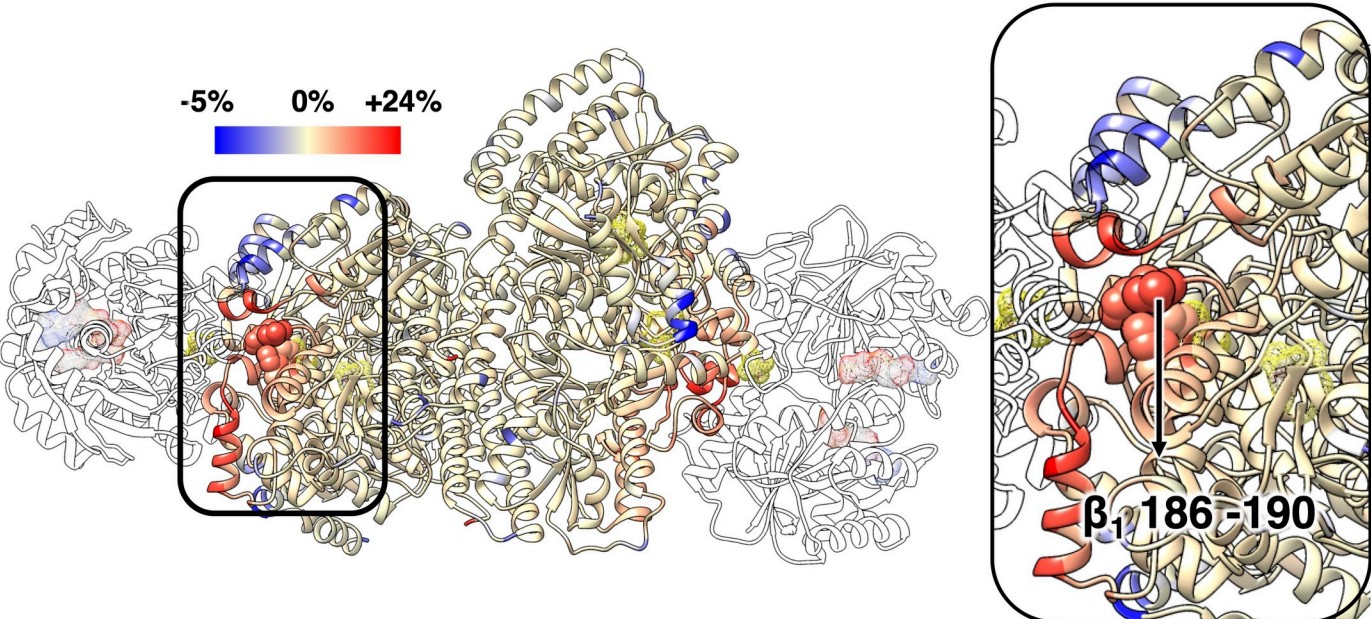

**Fig 4. Change in RMSF of the MoFe protein upon ATP hydrolysis.** The change is shown as deviation of the RMSF of the MoFe protein in the ADP-bound complex relative to the ATP-bound complex reported as a color scale and is projected on the ribbon representation of the protein. The positive changes, whereby the ADP-bound complex shows larger flexibility than the ATP-bound complex, are shown in red, and the negative values are shown in blue. For clarity of representation, the RMSF change for the Fe proteins is not color-coded. The ATP analogues, [4Fe-4S] clusters, P-clusters, and FeMo-co are shown in mesh surface. Residues 186–190 in $\beta_1$ are highlighted as space-filling structures (spheres). Residues at Fe protein/MoFe protein interface with large RMSF change upon ATP hydrolysis appear in orange or red.

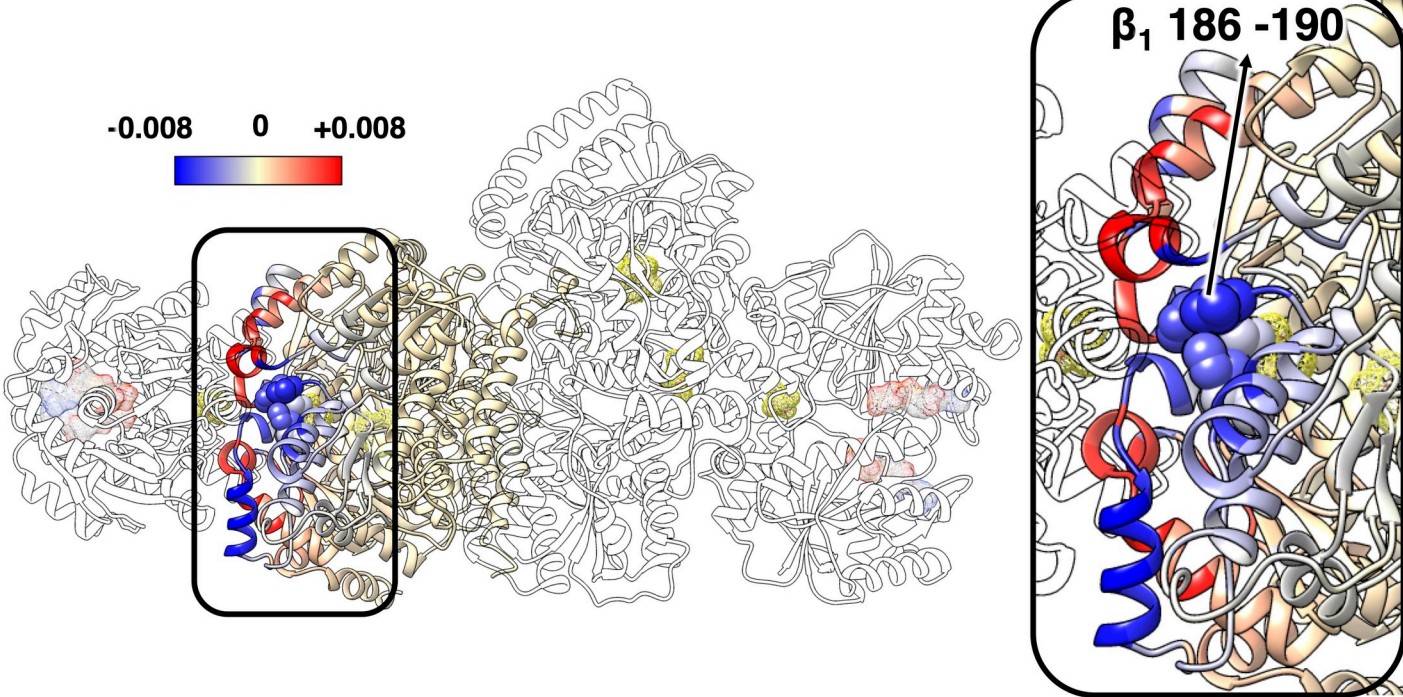

**Fig 5. Change in M between residues in $\alpha_1\beta_1$ to any residue in Fe$_1$ upon ATP hydrolysis.** The change is shown as largest deviation of **M** in the ADP-bound complex relative to the ATP-bound complex, either positive (red) or negative (blue) following the same projection scheme discussed in Fig 4. The Fe$_1$, Fe$_2$, $\alpha_2\beta_2$ proteins are shown in white. The ATP analogues, [4Fe-4S] clusters, P-clusters, and FeMo-co are shown in mesh surface. Residues 186–190 in $\beta_1$ are highlighted in spheres and residues 119–120 in $\alpha_1$ are indicated by arrow. Residues at Fe protein/MoFe protein interface with large deviation upon ATP hydrolysis appear in dark red or dark blue. For clarity of representation only the change in one half of the MoFe protein is shown.

the Fe protein/MoFe protein interface that are in close contact, longer range perturbations are also present. These perturbations extend up to the second coordination sphere of the P-cluster (Fig 4, inset). Interestingly, the largest of these perturbations involve the loop containing β-188$^{Ser}$, residues 186–190, which is located more than 10 Å from the closest residue of the Fe protein. Residues in this loop show an increase in the RMSF up to 16%. This increase should be compared to the maximum increase of about 24% for residues at the interface that lose contact with the Fe protein after the reorientation of the Fe protein on the MoFe protein upon ATP hydrolysis.

In addition to the differences in flexibility, the ATP- and ADP-bound complexes also present differences in the mechanical coupling. Some of these differences have already been highlighted above. Here, we further comment on additional differences at the Fe protein/ MoFe protein interface. The change in the mechanical coupling between Fe$_1$ and $\alpha_1\beta_1$ in the ADP- and ATP-bound complexes is illustrated in Fig 5. As can be seen, the mechanical coupling between the loop containing β-188$^{Ser}$ and the Fe protein is smaller in the ADP-bound complex. The decreased mechanical coupling and the increased flexibility of this loop in the ADP-bound complex are a consequence of less efficient coupling pathways (larger geodesic distance elements, $G_{ij}$) between the loop and the Fe protein.

In addition, the mechanical coupling between the Fe protein and the helices $\alpha_7$, $\alpha_8$ in the $\alpha$ component and the corresponding helices $\alpha_{33}$, $\alpha_{34}$ in the β component increase in the ADP-bound complex, except two residues at one end of the helix $\alpha_7$, $\alpha$-119$^{Gln}$ and $\alpha$-120$^{Glu}$, which are located near the β-188$^{Ser}$ loop. The other two helices ($\alpha_{35}$ and $\alpha_{36}$) in the β component, which are also located near the protein-protein surface, have decreased mechanical coupling

with the $Fe_1$ in the ADP-bound complex, while half of the corresponding helices $\alpha_9$ and $\alpha_{10}$ in the $\alpha$ component have increased. This asymmetric change of the mechanical coupling in the $\alpha$ and $\beta$ component may be due to the rolling motion of the Fe protein over the surface of the MoFe protein, from $\beta$ to $\alpha$ component (see below). This motion not only affects the protein-protein interaction at the interface but may also have influence on the mechanical coupling between the residues near the P-cluster and the Fe protein. The results between $\alpha_2\beta_2$ and $Fe_2$ are shown in S4 Fig.

## Communication network

The communication network between different regions of the nitrogenase complex was studied by analyzing the strongest mechanical coupling pathways between residues of one region and residues of another region as extracted from underlying mechanical coupling graph based on the most efficient paths described by the geodesic matrix. The residues with the largest influence on the mechanical coupling are identified by the cost-weighted betweenness centrality $\chi$. As we discussed in the Methods, the higher the $\chi$ value of a given residue, the higher the number of pathways passing through it; residues with large $\chi$ value represent effective chokepoints for the communication between two regions. We analyzed the communication within each half of the nitrogenase complex, and between the two halves of the nitrogenase complex. Results for the most important residues in the communication within each half of the complex (Fe-$\alpha\beta$), between the two halves (from $Fe_1$ to $Fe_2$), and between a specific region, the P-loop in the two halves (from P-loop in $Fe_1$ to P-loop in $Fe_2$) are show in Fig 6. In this figure, dominant $\chi$ values are reported as iso-surface enclosing the corresponding residues whose size is proportional to the magnitude of $\chi$.

**Communication within each half of the nitrogenase complex.** The most important residues of the communication network within each half of the nitrogenase complex are shown in Fig 6A as hot pink surfaces, whose size is proportional to $\chi^{Fe-\alpha\beta}$. It is remarkable that the $\chi^{Fe-\alpha\beta}$ values distribution is rather narrow. Only a few residues have a $\chi^{Fe-\alpha\beta}$ values close to the highest value. Out of 1532 residues of each Fe-$\alpha\beta$ half, only 7 are within 25% of the maximum $\chi^{Fe-\alpha\beta}$ value, 18 are within 50% and 58 are within 75% (S5A Fig, blue panel). The residues with the highest $\chi^{Fe-\alpha\beta}$ belong to a rather localized portion of the complex that extend from the switch regions in the Fe protein to the FeMo-co and include areas surrounding the [4Fe-4S] cluster and the P-cluster. Specifically, the residues near the Fe/$\alpha\beta$ interface (in particular $\beta$-183$^{Phe}$, $\alpha$-50$^{Lys}$, $\alpha$-164$^{Glu}$) have the largest influence (~90%). In addition, residues $\alpha$-63$^{Ala}$ and $\beta$-90$^{His}$, which are next to three residues that have been found important for the electron transfer ($\alpha$-64$^{Tyr}$, $\beta$-98$^{Tyr}$ and $\beta$-99$^{Phe}$), and the residue $\beta$-189$^{Phe}$ linked to the $\beta$-188$^{Ser}$ are found also important for the communication (40%, 45%, and 54%, respectively). Equally interesting, some of the residues in the FeMo-co binding pocket show $\chi^{Fe-\alpha\beta}$ values within the 50% of the highest $\chi^{Fe-\alpha\beta}$ value. Notably, these residues include $\alpha$-71$^{Val}$, which has been suggested to play a role in substrate selectivity.[47]

**Communication between the two halves of the nitrogenase complex.** The most important residues of the communication network between the two halves of the nitrogenase complex are shown in Fig 6B. Similar to $\chi^{Fe-\alpha\beta}$, the distribution of the $\chi^{Fe1-Fe2}$ values is also sharp (*i.e.*, 13, 39, and 147 residues out of 3064 are within the 25%, 50% and 75% of the maximum $\chi^{Fe1-Fe2}$ value, respectively; see also S5B Fig, pink panel). More specifically, the most important chokepoints are residues near the P-cluster (residue $\beta_1$-69$^{Ala}$, next to one P-cluster cysteine ligand $\beta_1$-70$^{Cys}$), the $\alpha_1\beta_1/\alpha_2\beta_2$ interface (residues $\beta_2$-479$^{Leu}$ and $\beta_2$-497$^{Leu}$), and $\alpha_2\beta_2/Fe_2$ interface (residue $Fe_{2B}$-111$^{Glu}$) (~90%). It is worth noting that the secondary structure element containing $\beta$-188$^{Ser}$ is also important for this communication, especially residues $\beta$-190$^{Val}$ and

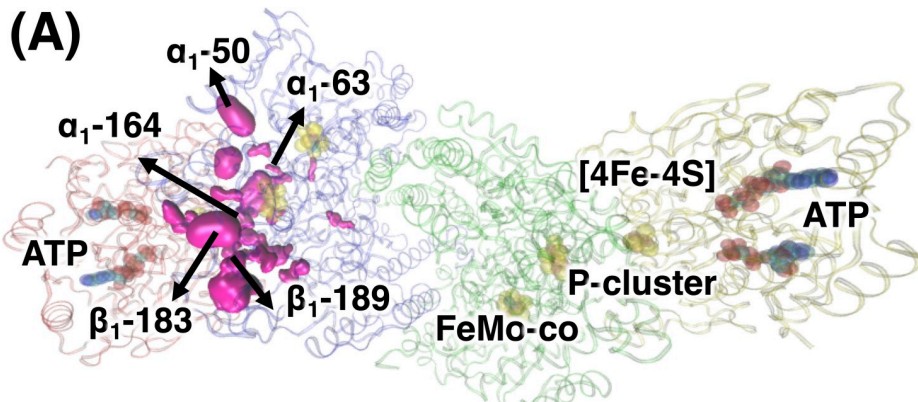

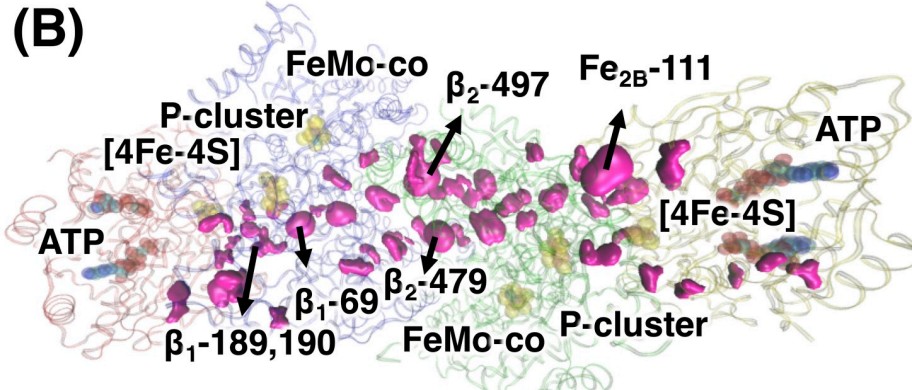

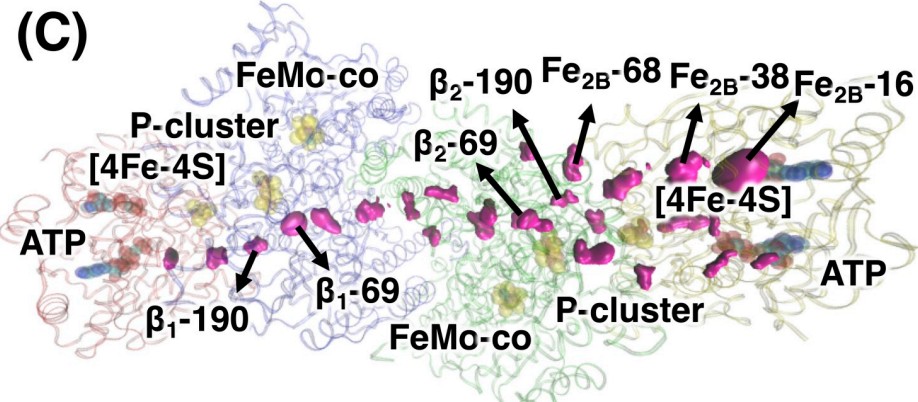

**Fig 6. Important residues for mechanical coupling between (A) $Fe_1$ and $\alpha_1\beta_1$, (B) $Fe_1$ and $Fe_2$ and (C) P-loops in $Fe_1$ and P-loops in $Fe_2$ as identified by cost-weighted betweenness centrality, $\chi$.** For clarity, only residues with $\chi >$ 40% of maximum value are shown as hot pink space-filling structures, whose size is proportional to the magnitude of the corresponding $\chi$ value. The ribbon presentation of the ATP-bound nitrogenase complex is colored as follows: $Fe_1$, red; $Fe_2$, yellow; $\alpha_1\beta_1$ unit of MoFe, blue; $\alpha_2\beta_2$ unit of MoFe, green. ATP, [4Fe-4S] clusters, P-clusters and FeMo-co are shown in spheres. Residues among top 10% of $\chi$ or residues in/near the sites of interest are labeled.

β-189$^{\text{Phe}}$, with χ$^{\text{Fe1-Fe2}}$ values within the 52% and 47% of the maximum. The residues with the highest χ$^{\text{Fe1-Fe2}}$ value form a nearly continuous treading from the switch region of one Fe protein to the switch region of the other Fe protein, passing through the [4Fe-4S] clusters and the P-clusters. The large majority of the communication pathways go through the side of the P-cluster opposite to the FeMo-co, and none of them passes near the FeMo-co itself (Fig 6B).

Among all of the structural elements composing the Fe proteins, the P-loops stand out as potential allosteric sites for the communication between the two Fe proteins. The most important residues of the communication network between the P-loops of the two Fe proteins are shown in Fig 6C. One residue within the P-loops (out of 32 residues) in Fe$_2$, Fe$_{2B}$-16$^{\text{Ser}}$, has the largest χ$^{\text{P-loop1-P-loop2}}$, followed by Fe$_{2B}$-38$^{\text{Cys}}$, close to the two switch regions of the Fe$_2$, and β$_1$-69$^{\text{Ala}}$, close to the P-cluster, both with χ > 80% of the maximum. It is worth noting that the loop enclosing β-188$^{\text{Ser}}$ (notably residue β-190$^{\text{Val}}$) is also important for the communication, with a χ equal to the 58% the maximum value. In addition, residue Fe$_{2B}$-68$^{\text{Glu}}$, located at the protein-surface end of the switch II region, has an appreciable χ value (~51% of the maximum) as well. Again, the large majority of the communication pathways go through the side of the P-cluster opposite to the FeMo-co, and none of them passes near the FeMo-co itself.

**Mechanical coupling and conformational dynamics.**   The mechanical coupling between distant parts of the nitrogenase complex is regulated by the underlying conformational dynamics. Allosteric signaling is instantaneously distributed through the scaffold of the complex via protein normal modes. Here, we analyze the protein vibrational modes that influence the communication network the most.

The influence of a given normal mode $k$ on the mechanical coupling in the nitrogenase complex was quantified by the overlap $O_k^{\text{X−Y}}$ between the χ values from the full covariance matrix, $\chi_0^{\text{X−Y}}$, and after removing normal mode $k$, $\chi_k^{\text{X−Y}}$ (Eq 6). Results are better visualized through the complement to 1 of $O_k^{\text{X−Y}}$, $I_k^{\text{X−Y}}$ (Eq 7). It is expected that the large-amplitude, slow-frequency delocalized modes will be most important for the long-range coupling, *i.e.*, they will have the highest $I_k^{\text{X−Y}}$. The $I_k$ values for the coupling between Fe$_1$ and α$_1$β$_1$ and between Fe$_1$ and Fe$_2$ calculated for the top 50 normal modes are reported in S6 Fig. As can be seen, the first 9 modes (except mode 5 and/or 6) affect appreciably both the communication within each half of the nitrogenase complex or between the two halves. On the other hand, some individual modes, *e.g.*, mode 1, 2, 3, or 4 have larger influence on the communication between the two halves (Fig 7B; $I_k^{\text{Fe1-Fe2}} > 0.40$), while the influence is much smaller on the communication within each half of the complex (Fig 7A; $I_k^{\text{Fe-αβ}} < 0.25$). As expected, the communication between the two halves relies more on delocalized motions, while more localized motions between Fe and αβ can give rise to the coupling within each half of the complex.

The $I_k$ values reported above refer to the global coupling between the Fe proteins and/or the MoFe protein, *i.e.*, all residues-to-all residues. We have also explored how the conformational dynamics of the complex couple specific regions of the nitrogenase complex. Results are qualitatively similar to those obtained for the global coupling. As an example, the $I_k^{\text{Ploop1-Ploop2}}$ for the coupling between the P-loops in the two Fe protein are given in Fig 7C.

The analysis discussed so far is based on harmonic vibrational modes, therefore on harmonic displacements about the equilibrium (crystallographic) configuration. Therefore, it cannot address the effect of transient structural changes, which can either enhance or weaken the relevance of the coupling pathways and, consequently, change the χ value associated with a given residue or region. More importantly, potential transient large-amplitude structural changes, postulated to be relevant for the electron transfer, are clearly not accessible through a Gaussian model. The only available information about the structural dynamics (change) during catalysis comes from the inspection of the crystal structure of the ATP- and ADP-bound

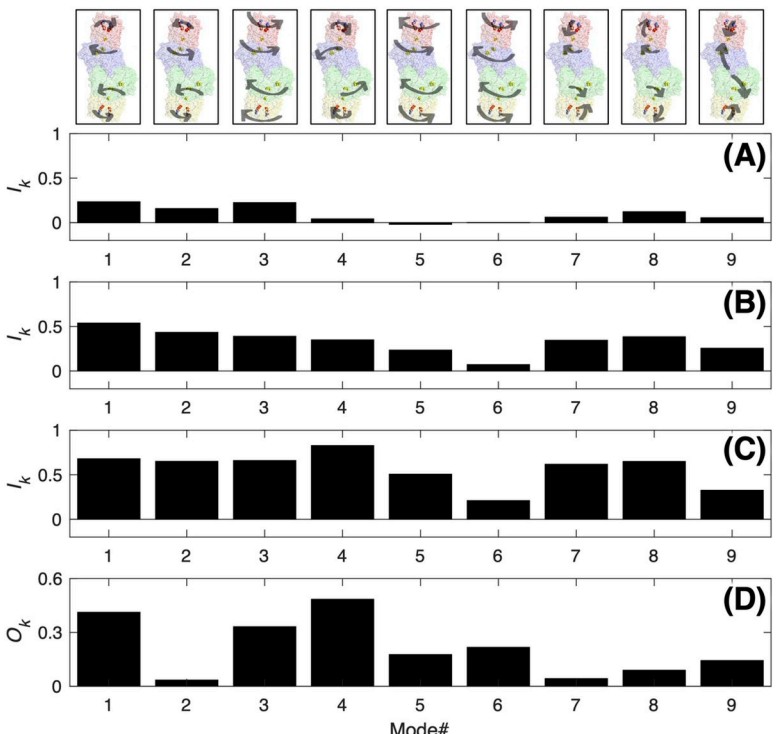

**Fig 7. Normal mode decomposition.** Top 9 normal modes of the ATP-bound nitrogenase complex (top panels) and their influences on the mechanical coupling between (A) $Fe_1$ and $\alpha_1\beta_1$, (B) $Fe_1$ and $Fe_2$, (C) P-loops in $Fe_1$ and P-loops in $Fe_2$; (D) overlap between the normal modes of the ATP-bound nitrogenase and the vector describing the displacement of the Fe protein that takes place upon ATP hydrolysis as inferred from the ATP- and ADP-bound crystal structures.

nitrogenase complex, which reveals a structural, rigid body rearrangement of the Fe protein over the MoFe protein upon ATP hydrolysis. It is therefore of interest to see if this structural reorganization correlates with the mechanical coupling deduced from equilibrium structures. To this end, the overlap between the vibrational modes and the unit vector, $\Delta\mathbf{R}$, describing the conformational change from the ATP-bound to the ADP-bound Fe protein complex is calculated. As can be seen from Fig 7D, modes 1, 3 and 4 account for majority of the structural rearrangement occurring upon ATP hydrolysis, with mode 4 having the largest overlap with $\Delta\mathbf{R}$. Remarkably, these three modes are among the most important modes for the mechanical coupling, especially between the P-loops of two Fe proteins.

Taking a step further, we investigated what type of mechanical coupling the displacement described by only the vector $\Delta\mathbf{R}$ would yield. In order to achieve this, the mechanical coupling analysis was repeated using a pseudo-"covariance matrix" constructed as $C'_{ij} = \Delta R_i \Delta R_j$ (S7 Fig). Various mechanical coupling pathways are examined, including those between the sites of interest: P-loops, [4Fe-4S] cluster, P -cluster, FeMo-co, switch I and II regions, the loop containing residues 186–190 in the β component of the MoFe protein. Results are reported in S8 Fig. Small asymmetries in the crystallographic structures are amplified in this analysis and result in an asymmetric pseudo-"covariance matrix", and consequently asymmetries in S8 Fig. It is clear that the motion described by the vector $\Delta\mathbf{R}$ can introduce an efficient mechanical coupling between the switch region I and the [4Fe-4S] cluster, the β-Ser[188] loop and the P-cluster within either half of the ATP-bound complex. Asymmetry between the two halves is observed since the mechanical couplings between the P-loop and β-Ser[188] loop, the P-cluster

and the FeMo-co within one half (*i.e.*, $Fe_1-\alpha_1\beta_1$) are similar to the ATP-bound complex itself, while the one between the [4Fe-4S] cluster and the P-cluster is similar within the other half (*i.e.*, $Fe_2-\alpha_2\beta_2$). Notably, the conformational motion taking place upon ATP hydrolysis can also introduce an efficient mechanical coupling between the loop containing β-Ser[188] of one half and the P-cluster of the other half of the ATP-bound complex.

### Testing of the mechanical coupling model

The computational model for the mechanical coupling discussed in the previous sections was validated against an analysis of the solution phase dynamics of the protein complex. We conducted a set of hydrogen-deuterium exchange experiments on the MoFe protein under two conditions, alone in solution and in the presence of ATP-bound Fe protein. Specifically, we monitored deuterium exchange after 1 hour, at which point all relevant long-time scale motions in the nitrogenase complex would have been sampled. In general, greater exchange was observed for peptides from free MoFe protein, than in the presence of ATP-bound Fe protein (Fig 8). Decreased exchange proximal to the MoFe protein/Fe protein interface was expected due to the burial of surface area and the formation of protein-protein interaction. What was not immediately expected were changes in deuterium uptake distal from the Fe protein binding site as well as in the protein core. The region showing the greatest exchange was near the P-cluster, for peptides containing residues β-188[Ser] and β-69[Ala]. In the presence of the Fe protein, the peptide with β-188[Ser] had a dramatic decrease in deuterium uptake, whereas the peptide with residue β-69[Ala] showed only a slight change. The changes detected in regions distal from the Fe protein docking site are consistent with our model of mechanical coupling.

To investigate the effects of the presence of Fe protein on long-distance communication of MoFe protein, we searched for regions with correlated patterns of H/D exchange. Based on the difference in percent deuterium exchange between regions, we built a correlation matrix for free MoFe protein and MoFe protein in the presence of ATP-bound Fe protein (S9 Fig). After

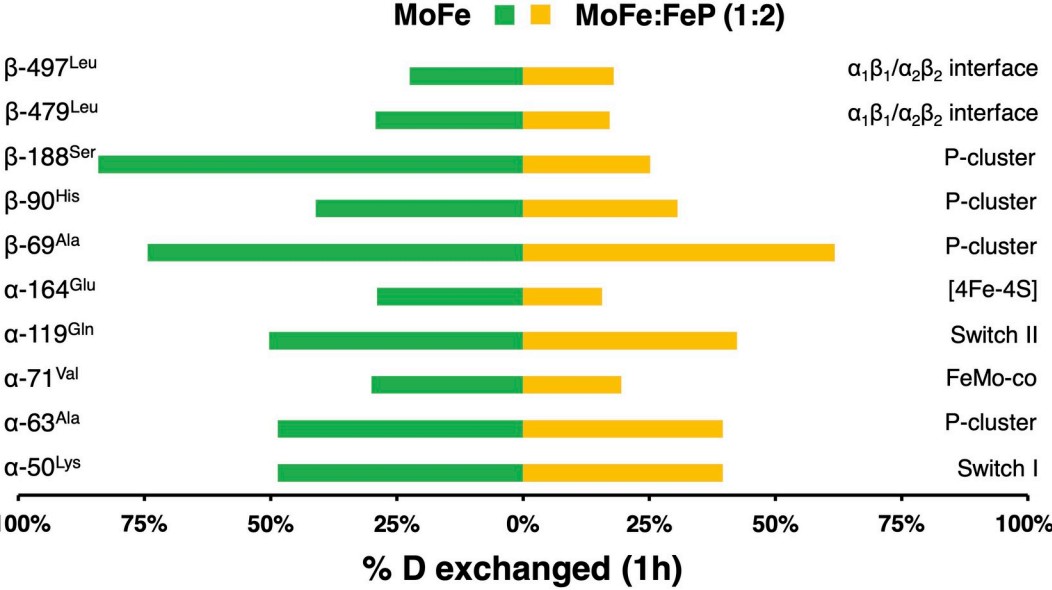

**Fig 8. HDX Mass Spectrometry measured deuterium exchange in the MoFe protein alone and in the presence of the ATP-bound Fe protein.** The percent deuterium uptake at 1 hour in selected regions of MoFe alone and in complex with Fe protein is shown as a histogram. Complete exchange was based on a 24-hour time point.

one hour of exchange, seven well-defined correlation groups were identified for each condition. However, the exchange patterns and peptides in the correlation groups were not the same between conditions. In MoFe protein alone, about 62% of investigated peptides show strong positive correlation (blue), while 22% have a strong negative correlation (red). In the presence of ATP-bound Fe protein, MoFe protein exhibits much greater negative correlation (36%). In MoFe alone, only peptides in the fifth cluster (green bars) have significant negative correlation. These peptides are spread across structure including the MoFe protein dimer interface, near Fe protein binding site, P-cluster and FeMo-co. Enhanced negative correlation upon binding of Fe protein is consistent with a model of negative cooperativity between "halves" of MoFe protein.

To evaluate our mechanical coupling model, we investigated the correlation patterns of the "chokepoint" residues. On the experimental time scale, H/D exchange revealed four types of correlation patterns in free MoFe protein and in the presence of the ATP-bound Fe protein (Fig 9). The first group contains the following residues: $\alpha$-50$^{Lys}$ (peptide 39–54 near Switch I region of the Fe protein), $\alpha$-63$^{Ala}$ (peptide 63–85 near the P-cluster) and $\alpha$-119$^{Gln}$ (peptide 119–134 near Switch II region of the Fe protein). The second group contains the following residues: $\alpha$-71$^{Val}$ (peptide 67–85 near the FeMo-co), $\alpha$-164$^{Glu}$ (peptide 151–165 near the [4Fe-4S] cluster in the Fe protein), $\beta$-479$^{Leu}$ and $\beta$-497$^{Leu}$ (peptide 473–489 and 495–523 at the $\alpha_1\beta_1$-$\alpha_2\beta_2$ interface). Both groups show positive correlation with H/D exchange in the free MoFe protein. When the full nitrogenase complex is formed, trends in protein dynamics for group one and two become negatively correlated. The third and fourth groups contain residues in the $\beta$ subunit that are near the P-cluster, $\beta$-69$^{Ala}$ (peptide 58–77), $\beta$-90$^{His}$ (peptide 78–98) and $\beta$188$^{Ser}$ (peptide 183–192) and correlation patterns are dependent on complex stoichiometry. In the free MoFe protein, the residues $\beta$-69$^{Ala}$ and $\beta$-188$^{Ser}$ are strongly correlated with each other and negatively correlated with $\beta$-90$^{His}$ as well as with the first and second group. In the ATP-bound Fe protein, these three residues are part of a more nuanced interaction network.

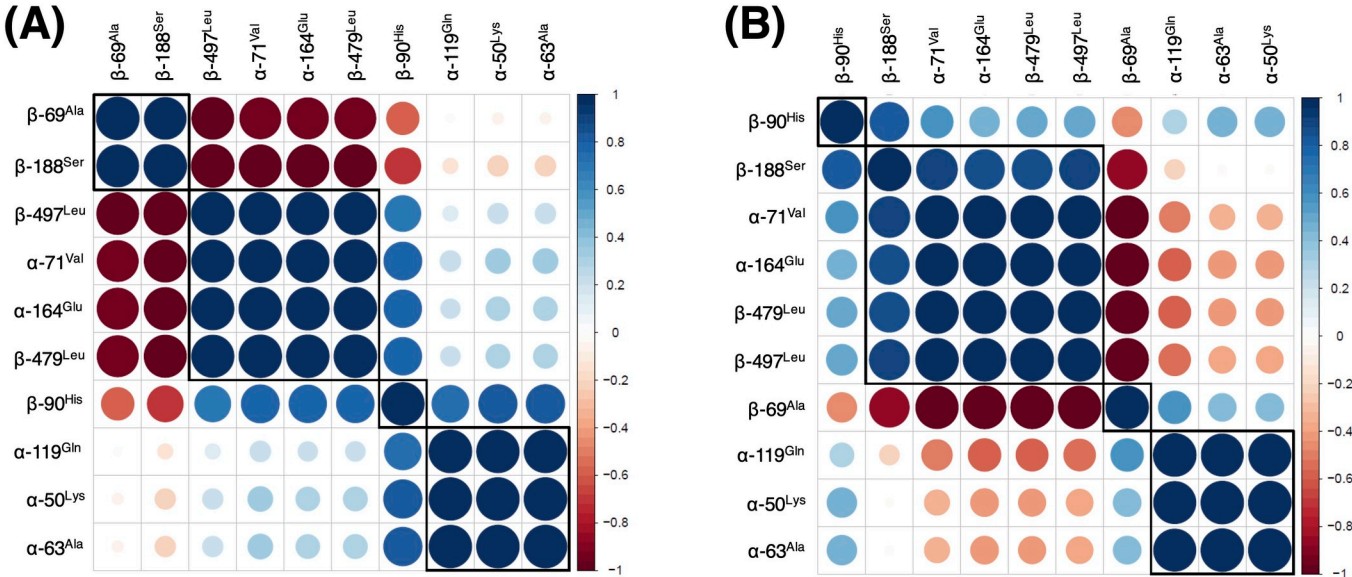

**Fig 9. Correlation patterns of protein dynamics in (A) free MoFe protein and (B) MoFe protein in presence of ATP-bound Fe protein based on HDX measurements recorded after 1h of the deuterium exchange.** Red and blue circles correspond to negative and positive correlations, respectively. An empty spot in the matrix (or white circle) signifies lack of any correlation. Size of the circle represents significance of the results: the larger the circle the more significant the correlation. Squares are drawn to highlight clusters with similar correlation patterns.

For example, the region around residue β-90$^{His}$ remains similarly correlated with all peptides except the one containing residue β-188$^{Ser}$ (which changes from negative to positive). Whereas the peptide containing β-188$^{Ser}$ has positive correlation with all residues from the second group and negative correlation with β-69$^{Ala}$. The peptide containing β-69$^{Ala}$ changed from positive to negative correlation with β-188$^{Ser}$ and from negative to positive correlation with group one upon Fe protein binding.

Detailed analysis of the theoretical and experimental data made it immediately evident that regions identified as important for the Fe protein/MoFe protein coupling in the computational analysis correspond (or are adjacent) to regions with high H/D exchange correlation. In addition, the analysis revealed that the region surrounding the P-cluster is sensitive to the presence of ATP-bound Fe protein. This agrees with observed structural changes when nitrogenase is poised at different steps in the catalytic cycle.[28] Of particular interest is cross-talk between β-69$^{Ala}$ and β-188$^{Ser}$, two resides proximal to the P-cluster ligation site. In free MoFe protein, these residues are positively correlated. Upon binding of ATP-bound Fe protein, β-69$^{Ala}$ and β-188$^{Ser}$ (along with the P-cluster environment in the β subunit) become negatively correlated. At the same time, H/D exchange of β-69$^{Ala}$ is positively correlated with the P-cluster environment in the α subunit of the MoFe protein. Equally interesting is the relationship of β-188$^{Ser}$/β-69$^{Ala}$ protein dynamics with α$_1$β$_1$-α$_2$β$_2$ interface (residues β-479$^{Leu}$/β-497$^{Leu}$). Despite spatial separation, the P-cluster ligation site and the α$_1$β$_1$/α$_2$β$_2$ interface show a strong correlation. However, only β-188$^{Ser}$ is sensitive to the presence of ATP-bound Fe protein. In summary, our H/D exchange experiments reveal correlated patterns of dynamics precisely where computationally derived mechanical coupling networks predict.

## Conclusions

We analyzed the mechanical coupling in the nitrogenase complex using a graph theory-based approach which builds on the covariance matrix **C** of the displacements of amino acid residues as obtained from an anisotropic network model.[38] Hydrogen/Deuterium (H/D) exchange experiments were performed to provide a validation of the theoretical predictions.

The covariance matrix shows positive correlation between the motion of the two halves of the complex and negative correlation of the motion within each half (*i.e.*, Fe$_1$ and α$_1$β$_1$) in general. The root mean-square fluctuation (RMSF) of residues of the MoFe protein complexed with ATP-bound Fe protein is generally smaller than that of the MoFe protein complexed with the ADP-bound Fe protein, suggestive of increased rigidity in the complex with ATP-bound Fe protein. In particular, we found that the residues in the loop containing β-188$^{Ser}$ show a peculiar dynamical behavior. Residue β-188$^{Ser}$ has been proposed to act as a rectifier, which allows a unidirectional flow of electrons from the Fe protein to the MoFe protein.[35,48] In addition, the mechanical coupling between this loop and the Fe protein is stronger in the ATP-bound complex. During each electron delivery cycle, an electron is delivered from the P-cluster to the FeMo-co (oxidation of the P-cluster), and then from the Fe Protein to the P-cluster (reduction of the P-cluster). Available crystal structures of the MoFe protein with the P-cluster in different oxidation states suggest that, upon oxidation, the P-cluster binds β-188$^{Ser}$, and then upon reduction and ATP hydrolysis, the β-188$^{Ser}$ dissociates from the P-cluster. This is confirmed in computational analysis that the one-electron oxidized P-cluster is coordinated by the hydroxyl of β-188$^{Ser}$, while the reduced P-cluster is not.[35] Our analysis (based on the AMPPCP-bound crystal structure) shows that in the ATP-bound complex the loop containing β-188$^{Ser}$ has a decreased RMSF with respect to the isolated MoFe protein, while the opposite behavior is observed for the ADP bound complex. Regardless of whether the β-188$^{Ser}$ is directly involved in gating the electron transfer, or it stabilizes the P-cluster after it loses the

electron as suggested recently for the wild MoFe protein,[36] having a suitable protein environment near β-188$^{Ser}$ seems to be advantageous. In fact, although the exact role of β-188$^{Ser}$ is still not clear, the dynamics of position β-188$^{Ser}$ can be highly dependent on the residues nearby, which appear to play an important role in allosteric communication. Therefore, mutation of residue β-188$^{Ser}$ is not expected to shut down these pathways.

The mechanical coupling between various regions of the nitrogenase complex was investigated through the mechanical coupling matrix **M**, calculated by weighting the covariance of the displacements between each pair of residues by the geodesic distances between them. The geodesic distance is defined as the sum of all edge lengths along the shortest path from the residue $i$ to residue $j$ as obtained from the matrix **C**. The matrix **M** indicates communication within each half and between the two halves of the complex. We like to emphasize that correlation or communication here does not imply causality. In fact, strong mechanical coupling indicates an efficient coupling pathway connecting two residues or regions which show correlated or anti-correlated motions. Clearly, the existence of a pathway is a necessary but not sufficient condition for causality. The study of the effect of suppression or delay of the motion of one region on the motion of the other,[49] and/or the analysis of the energetics of allosteric signaling[50–52] have been recently proposed to address causality of the correlation.

The strength of the coupling over the ADP-bound complex is noticeably larger than the ATP-bound complex, which may not suggest a more efficient network of communication after ATP hydrolysis but a less compact and thus more flexible structure of ADP-bound complex. Within one half of the complex, the loop containing β-188$^{Ser}$ is coupled to the switch regions of the Fe protein in the ATP-bound complex but not in the ADP-bound complex. This observation further confirms a communication between the MoFe protein and the Fe protein, importantly, that the ATP-bound Fe protein can allosterically modulate the motion of the β-188$^{Ser}$ loop.

Consistent with previous proposals based on structural analysis only, the present analysis suggests that the P-loops located near the nucleotide binding sites in the Fe protein are also important for allosteric communication. In particular, we found that not only are they involved in the communication within each half, but they are central for the communication between the two halves of the nitrogenase complex. Indeed, we found that the P-loops and other surrounding residues at the nucleotide binding site in one Fe protein are mechanically coupled with those of the other Fe protein, over 130 Å apart. Even if our analysis cannot provide information on the resulting effect of this mechanical coupling, it suggests that a perturbation near the P-loop on one end of the complex (for instance hydrolysis of ATP or release of inorganic phosphate, Pi) is propagated to the other end, thereby having a potential regulatory effect.

The strongest mechanical coupling pathways between the two Fe proteins (all residues of Fe$_1$ vs. all residues of Fe$_2$) pass through residues near the P-cluster but not the FeMo-co. Although this analysis does not provide information on the role of this mechanical coupling between the two Fe proteins, it underscores the importance of the residues around the P-cluster in the long-range communication. It is widely believed that the protein environment transiently changes in order to initiate the electron transfer,[23] and residues near the P-cluster are likely playing an important role in the electron transfer process.

The importance of individual residues or structural elements for the coupling between distant parts of the complex was quantified in terms of the cost-weighted betweenness centrality, χ. A large value of χ for a given residue or region indicates that a large number of pathways between those distant regions pass through that residue, and consequently that that residue is highly important for the communication. Only a few residues, either near the protein-protein interface or adjacent to residues experimentally proven important for the electron transfer have large χ. This suggests that, while the allosteric communication network is resulted from a

large number of pathways, they share a few common key residues (nodes). These residues represent chokepoints for the allosteric communication either within the nitrogenase subunits or in the long-range inter-subunit communication. The limited number of those residues makes them ideal target for mutagenesis study to assess their potential role in regulating the function of nitrogenase.

In a simple harmonic description, protein dynamics are the results of the superposition of vibrational modes about an equilibrium position. We analyzed the influence of each mode on the mechanical coupling between different regions of the nitrogenase complex. Motions introduced by an individual mode are insufficient to form an efficient mechanical coupling pathway between two distant regions of the complex: removing just one mode will degrade but not completely shut down the communication. In other words, the covariance matrix obtained by removing only one mode still provides a fully connected graph. However, we found that large-amplitude, slow-frequency motions are far more important than the small-amplitude, fast motions; it can be difficult to well converge the former in the MD simulations. In particular, the rolling motion of the Fe protein over the MoFe protein has the largest influence on the mechanical coupling between the P-loops of the two Fe proteins (modes 1, 3 and 4, Fig 7). These modes also yield the largest overlap with the vector describing the conformational change from the ATP- to ADP-bound nitrogenase as evinced from their crystal structures. This suggests that conformational changes near the switch regions of the Fe protein initiated from the P-loops upon ATP hydrolysis (as deduced from inspection of available crystal structures) can induce a response in the other half of the MoFe protein and, remarkably, in the other Fe protein.

H/D exchange experiments provided an important validation for the mechanical coupling network between the Fe protein and the MoFe protein. Comparison between the uptake of deuterium in the MoFe protein only and the ATP-bound nitrogenase complex supported the profound influence of the Fe protein in the dynamics of the MoFe protein. These measurements showed that when two Fe proteins are bound to the MoFe protein, the H/D exchange is appreciably reduced. The decrease goes beyond the interface between the MoFe protein and Fe protein and extends to inner parts of the MoFe protein identified computationally as mechanically coupled to the Fe protein (or nearby these regions). Specifically, correlations in the H/D exchange data highlighted four groups of the H/D exchange patterns: (i) residues in the $\alpha$ subunit in the P-cluster environment, near the switch I and II regions of the Fe protein; (ii) residues in the FeMo-co environment, near the [4Fe-4S] cluster of the Fe protein and the $\alpha_1\beta_1/\alpha_2\beta_2$ interface; (iii) and (iv) residues in the $\beta$ subunit in the P-cluster environment. The correlation in the H/D exchange indicates a coupling in the dynamics of those peptides. The change in correlation pattern from the free MoFe protein to the nitrogenase complex suggests that the motions of the residues in the four groups above are largely influenced by binding two Fe proteins. Residues highlighted in the H/D exchange either show strong mechanical coupling directly to the Fe protein or appear to be the chokepoints on the mechanical coupling pathways from the Fe protein to the MoFe protein or to the other Fe protein, which also suggests their importance in the protein-protein communication.

Although the present analysis does not yet provide sufficient information to directly derive a step-by-step regulatory mechanism from the observed mechanical coupling in the nitrogenase complex, it is able to identify structural motifs that have been proposed experimentally to be important for the enzymatic regulation as the most strongly coupled. Importantly, it allows quantifying the magnitude of the coupling between any two residues or two regions, and furthermore identifying potential allosteric pathways between them. In particular, our analysis informs us how the two halves of nitrogenase might communicate and what the most important regions for this communication are, laying down the basis for future mechanistic studies.

## Supporting information

**S1 Table. Crystal structures of the ADP- and ATP-bound nitrogenase complexes.**
(DOCX)

**S1 Fig. Difference matrix between the mechanical coupling matrices M of ADP-bound complex and the ATP-bound complex ($M_{ADP}$-$M_{ATP}$).**
(TIF)

**S2 Fig. Normalized correlation matrix C of the residues displacements for (A) the ATP-bound nitrogenase complex and (B) the ADP-bound nitrogenase complex.** The correlation between $Fe_1$ and $\alpha_1\beta_1$, $Fe_2$ and $\alpha_2\beta_2$, $Fe_1$ and $Fe_2$ are highlighted in dashed blocks. The locations of the cysteine ligands to the [4Fe-4S] cluster in $Fe_1$ and P-cluster in $\alpha_1\beta_1$ are indicated by arrows in correlation between $Fe_1$ and $\alpha_1\beta_1$.
(TIF)

**S3 Fig. Maximum mechanical coupling M.** (A) residues in $\alpha_1\beta_1$ to any residue in $Fe_1$ and (B) residues in $Fe_2$ to any residue in $Fe_1$ in ADP-bound complex. The ATP analogues, [4Fe-4S] clusters, P-clusters, and FeMo-co are shown in mesh surface. Residue 186–190 in $\beta_1$ are shown in spheres in (A); one $\alpha$-helix and three $\beta$-sheets in each monomer of $Fe_2$ appear red in (B) (following the numbering in the 2AFI [I] PDB file).
(TIF)

**S4 Fig. Change in M of between residues in $\alpha_2\beta_2$ to any residue in $Fe_2$ upon ATP hydrolysis.** The change is shown as largest deviation of **M** in the ADP-bound complex relative to the ATP-bound complex, either positive or negative. The $Fe_1$, $Fe_2$, $\alpha_2\beta_2$ proteins are shown in white. The ATP analogues, [4Fe-4S] clusters, P-clusters, and FeMo-co are shown as mesh surfaces. Residues 186–190 in $\beta_1$ are highlighted in spheres. Residues at Fe protein/MoFe protein interface with large deviation upon ATP hydrolysis appear in dark red or dark blue.
(TIF)

**S5 Fig. $\chi^{X\text{-}Y}$ of residues in the ATP-bound complex.** (A) X = $Fe_1$ and Y = $\alpha_1\beta_1$; (B) X = $Fe_1$ and Y = $Fe_2$. Each circle represents a residue. $\chi$ is shown as the percentage relative to the corresponding maximum $\chi^{X\text{-}Y}$.
(TIF)

**S6 Fig. $I_k^{X\text{-}Y}$ for the first top 50 normal modes for mechanical coupling.** (A) X = $Fe_1$ and Y = $\alpha_1\beta_1$; (B) X = $Fe_1$ and Y = $Fe_2$, (C) X = P-loops in $Fe_1$ and Y = P-loops in $Fe_2$.
(TIF)

**S7 Fig. Normalized pseudo-covariance matrix constructed from the displacement of the ATP- to ADP-bound complex.**
(TIF)

**S8 Fig. Comparison of the coupling pathways.** Overlaps of coupling pathways between structural elements X and Y, *i.e.*, $O^{X\text{-}Y}$, between the motions of the ATP-bound complex as obtained from ANM and the ATP- to ADP-bound complex atomic displacement as inferred from the corresponding crystal structures are shown. The subscript of the X or Y elements indicates which half of the complex the element belongs to.
(TIF)

**S9 Fig. Correlation patterns of peptide-level hydrogen-deuterium exchange in free MoFe protein (top) and MoFe:FeP(ATP) 1:2 (bottom) measured at 1 hour of exchange.** Red and blue circles correspond to negative and positive correlations, respectively. White indicates no

correlation. Diameter of the circle represents significance; larger is more significant correlation. Peptides are group based on hierarchical clustering. Green bar indicates peptides which are negatively correlated in free MoFe protein.
(TIF)

## Author Contributions

**Conceptualization:** Qi Huang, Lewis E. Johnson, Simone Raugei.

**Data curation:** Qi Huang, Monika Tokmina-Lukaszewska, Hayden Kallas.

**Formal analysis:** Monika Tokmina-Lukaszewska.

**Funding acquisition:** John W. Peters, Lance C. Seefeldt, Brian Bothner, Simone Raugei.

**Investigation:** Qi Huang, Monika Tokmina-Lukaszewska, Lewis E. Johnson, Simone Raugei.

**Methodology:** Qi Huang, Lewis E. Johnson, Simone Raugei.

**Project administration:** Simone Raugei.

**Resources:** John W. Peters, Lance C. Seefeldt, Brian Bothner.

**Supervision:** Bojana Ginovska, John W. Peters, Lance C. Seefeldt, Brian Bothner, Simone Raugei.

**Validation:** Qi Huang, Monika Tokmina-Lukaszewska.

**Visualization:** Qi Huang, Monika Tokmina-Lukaszewska.

**Writing – original draft:** Qi Huang, Simone Raugei.

**Writing – review & editing:** Qi Huang, Monika Tokmina-Lukaszewska, Lewis E. Johnson, Bojana Ginovska, John W. Peters, Lance C. Seefeldt, Brian Bothner, Simone Raugei.

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
