## [Decision Letter · Decision Letter 0]

22 Sep 2020

Dear Dr Raugei,

Thank you very much for submitting your manuscript "Mechanical Coupling in the Nitrogenase Complex" for consideration at PLOS Computational Biology.

As with all papers reviewed by the journal, your manuscript was reviewed by members of the editorial board and by several independent reviewers. In light of the reviews (below this email), we would like to invite the resubmission of a significantly-revised version that takes into account the reviewers' comments. Towards this end, a stronger connection and discussion of the results of calculations with the HDX experiment is absolutely needed. By the way, our recent publication (https://doi.org/10.1016/j.str.2020.03.014; on a very different system) may help you address Reviewer 1's comment about causality.

We cannot make any decision about publication until we have seen the revised manuscript and your response to the reviewers' comments. Your revised manuscript is also likely to be sent to reviewers for further evaluation.

Sincerely,

Turkan Haliloglu

Associate Editor

PLOS Computational Biology

Nir Ben-Tal

Deputy Editor

PLOS Computational Biology

Reviewer's Responses to Questions

**Comments to the Authors:**

Reviewer #1: Allosteric regulation in the nitrogenase complex is studied in this work by means of graph theory analysis. The motivation for this work is the lack of complete understanding of the mechanism of allosteric communication at different stages of the protein functional cycle. There are partial data on local conformational changes and indirect evidences on the effect of mutations on functional sites of the complex. All the above prompted authors to explore the mechanical coupling between the protein sites, using graph theoretical approach developed in the group.

The authors demonstrate “mechanical coupling” between functionally involved regions, as well as analyze significance of individual normal modes contributions to the coupling. They also support their calculations with the data from HDX experiments. The major caveat of the approach is that it is based on the analysis of correlated motions, while correlation does not necessarily causality. Further, they claim that regions with “highly correlated or anti-correlated motions are mechanically coupled through efficient energy transfer path”. This conclusion seem to be unjustified as in the condensed matter of the protein there are no selected channels of the energy transformation, and all work produced in the structure is a result of multibody interactions between amino acid residues. If authors should resort to network consideration of protein, they should not be mixing it with the claims on the analysis of the protein energetics. If they decide to consider the causality and energetics, they may use, for example, AlloSIgMA web-server (PMID: 32392302, PMID: 29106449), allowing to calculate the allosteric free energy as a result of the perturbationand to estimate effects of mutations (PMID: 26939022, PMID: 30827842).

The HDX data is of interest, but should be better linked to results of calcualtions and further discussed.

The discussion is too lengthy, sometimes too speculative when results on mechanical coupling are interpreted in terms of the energy/electron transfer.

“Important allosteric centers” is a bad phrasing.

Figures 2 and 3 should be merged in one, or Figure 2 should be moved to supplementary. Figure 9 - to supplementary as well.

Reviewer #2: The authors describe an engaging and comprehensive study of synchronized and allosteric motion of the nitrogenase proteins. Importantly, the theoretical studies are connected to isotope effect experiments, and the entire body of work is place into an appealing modern perspective. While this form of modeling is simple and fast, it can be extremely useful and I believe that the paper will prove to be of great value and interest to researchers in the field. There was an early study of Liao and Beratan that reported coarse grained normal mode analysis of the nitrogenase proteins as well (see Biophysical Journal 87 1369–1377 (2004)). It would be interesting to compare the results of the two studies and to place the early albeit less comprehensive study into the context of this modern analysis, if possible.

Reviewer #3: Huang et al. identified communication pathways in the nitrogenase complex comprised of two Fe and two FeMo-co proteins. Nucleotide hydrolysis in the Fe- protein initiates a long-range electron transfer from the Fe-proteins to the FeMo-co. Electron transfer to the active site of FeMo occurs by intermediate electron transfer steps through the 4Fe-4S clusters and P-cluster. It is known (also mentioned by the authors) that the nucleotide hydrolysis and the consequent electron transfer is coupled with some structural rearrangements, especially at the interface of the Fe and two FeMo-co proteins. Structures of the complex before and after the hydrolysis are available. These two endpoints indicate that the structural changes may involve a rigid body movement of the Fe-proteins. Additionally, electron transfer in one subunit slows down the electron transfer in the other subunit. The mechanism of communication is unknown. The authors performed a sophisticated network analysis using just these two structures as inputs to unravel the mechanistic pathway.

The paper presents a very sophisticated analysis for an input of just two structures. However, in my experience, it is often the case that pathways connecting two endpoint structures require further optimization as local dynamics may enhance or diminish parts of the path. Can the authors comment on this possibility?

Additionally, the 2AFK structure (used by the author) has been made obsolete and replaced in the PDB by 4WZB. Can the authors compare the structures to ensure something wasn't incorrect in the older, obsolete one?

The betweenness centrality used here relies on the shortest-path connecting distant domains. However, an allosteric path does not always take the shortest path through the protein systems. Sometimes, depending on specific residues, a slightly longer path may be more effective for communication, especially when electron transfer is involved. The effect of the distance can be offset by the exponential energetic dependence of the electron transfer rates. Do the authors agree with this possibility? If so, can they extend their analysis to consider pathways other than the shortest path?

I'm unclear on how the correlations were determined for the H/D experiments. Can the authors add a more detailed explanation along with an equation?

Other minor points:

Page 2: "800 Ang.^2,20" It's hard to read this because the formatting of the reference is identical to the exponent; please revise.

Page 17: "two show opposite trend in..." - trends

Figure 9: "Swtich" in the figure should be "Switch".

Reviewer #4: The manuscript proposed by Simone Raugei and co-workers deals with the investigation of the mechanical coupling in the Nitrogenase complex by means of a recent developed technique that the authors already presented in a previous publication (ref. 38 in the manuscript).

The paper is well written, and the result section is clear, maybe conclusion could be more concise. After minor revisions I recommend this manuscript for publication in PLOS computational biology.

Basically I have only some doubts from the technical part that I believe the authors will be able to dispel without problems.

If I have understood correctly, in the technique adopted by the authors one compute the covariance matrix for two crystal structures (those reported in Table S1 of the SI) reported in ref. 29. It is important here to underline for clarity some more features of these two pdb.

• Why in table S1 are reported different number of residues for 2AFK and 2AFI, if the enzyme is the same and come from the same organism (Azotobacter vinelandii)?

• These differences maybe are not significant for the analysis of the matrix C, but the authors should say something about…

• How many residues in the two pdb?

Regarding matrix C:

• Only Calpha are considered? Or more?

• I guess that all cofactors are considered, right?

Then, why the authors decided to consider this approach instead of a molecular dynamic (MD) approach? We all know that MD for such metal-protein systems is quite demanding, but a word on this issue should be beneficial for the readers.

Finally, the authors validated their results based on the H / D exchange experiments, a fact that gives value to the manuscript. However C matrix is computed on just two structures, and I would like that the authors could comment this in light of the standard MD PCA analysis made on a complete trajectory.

**Have all data underlying the figures and results presented in the manuscript been provided?**

Reviewer #1: Yes

Reviewer #2: None

Reviewer #3: None

Reviewer #4: Yes

PLOS authors have the option to publish the peer review history of their article (what does this mean?). If published, this will include your full peer review and any attached files.

Reviewer #1: No

Reviewer #2: No

Reviewer #3: No

Reviewer #4: No
---

## [Decision Letter · Decision Letter 1]

1 Dec 2020

Dear Dr Raugei,

Thank you very much for submitting your manuscript "Mechanical Coupling in the Nitrogenase Complex" for consideration at PLOS Computational Biology. As with all papers reviewed by the journal, your manuscript was reviewed by members of the editorial board and by several independent reviewers. The reviewers appreciated the attention to an important topic. Based on the reviews, we are likely to accept this manuscript for publication, providing that you modify the manuscript reflecting the review recommendations of Reviewer #1 as best as you can. 

Sincerely,

Turkan Haliloglu

Associate Editor

PLOS Computational Biology

Nir Ben-Tal

Deputy Editor

PLOS Computational Biology

[LINK]

Reviewer's Responses to Questions

**Comments to the Authors:**

Reviewer #1: The current revision is not satisfactory for reasons below and should be amended.

Authors:

Here we would like to reiterate that the method we have developed was designed to filter out

pairs of residues whose covariant motion is purely coincidental and evaluate potential

communication pathways between distant region of a protein complex. The major difference

between our M-matrix analysis and a simple covariance analysis of motion, is that if two residues have correlated motions but do not have an efficient path for mechanical coupling between them (large geodesic distance in the covariance space), a small (poorly coupled) mechanical coupling (as described by the M matrix) will be produced. In this way, geodesic weighting filters out motions that are merely coincidental. Clearly, the existence of a pathway is a necessary but not sufficient condition for causality. This is actually an important point, which was added in the revised manuscript (page 23).

Reviewer:

Explain directly in the work what “mechanical coupling between them (large geodesic distance in the covariance space)” means in terms of protein dynamics that underlies allosteric communication.

Authors:

“We like to emphasize that correlation or communication here does not imply causality. In fact, strong mechanical coupling indicates an efficient coupling pathway connecting two residues or regions which show correlated or anti-correlated motions. Clearly, the existence of a pathway is a necessary but not sufficient condition for causality. A study of effects of suppression or delay of the motion of one region on the motion of the other will be necessary to consider causality.”

Reviewer:

It should be directly stated in teh above paragraph that causality can only be quantified via analysing the enegetics of allosteric signalling with corresponding references (PMID: 26939022, PMID: 29912863, PMID: 30439587).

Authors:

We also realize that the language adopted in the manuscript was not clear. In particular, as

originally discussed, the concept of energy transfer could have led to misinterpretations. We used the term energy transfer to indicate instantaneous flow of kinetic (thermal) energy mediated by protein vibrations (normal modes). We have revised and consolidated the language to address this point. For example, where appropriate, “energy” is changed to “information” and “energy flow” changed to “communication”.

Reviewer:

“information”, “structural or dynamic information”, “information transfer” and other similar usage of the word “information” is not correct and should be changed throughput the text.

Authors:

There was an early study of Liao and Beratan that reported coarse grained normal mode analysis of the nitrogenase proteins as well (see Biophysical Journal 87 1369–1377 (2004)). It would be interesting to compare the results of the two studies and to place the early albeit less comprehensive study into the context of this modern analysis, if possible.

We thank the Reviewer for pointing out the work by Liao and Beratan. We compare their results with ours in the Results and Discussion of the revised manuscript (page 12).

“Within each Fe protein, the P-loops are coupled to the protein-protein interface and the [4Fe-4S] cluster through the switch region I and II, respectively. Liao and Beratan discussed this coupling in their computational analysis of the isolated Fe protein. Using a coarse-grained model, they identified correlated regions in the Fe protein and reported that residues in the P-loops and switch regions are relatively rigid in the slowest normal mode, which can be important for function-relevant conformational changes. As we will discuss below, we show that there is indeed an efficient communication pathway between these regions, which, in the nitrogenase complex, extends up to the interior of the MoFe protein.”

Reviewer:

Since authors compared their results with those od Liao and Beratan at the request of second reviewer, I do not understand why they did not obtain the picture of the energetics of allosteric signalling using AlloSigMA cited in my previous review. In this case authors would be able to talk about causality and energetics avoiding to use wrong terminology as noted above.

Reviewer #2: The authors have addressed all referee concerns and the paper is suitable for publication.

Reviewer #3: The authors have sufficiently addressed my concerns.

**Have all data underlying the figures and results presented in the manuscript been provided?**

Reviewer #1: Yes

Reviewer #2: None

Reviewer #3: None

PLOS authors have the option to publish the peer review history of their article (what does this mean?). If published, this will include your full peer review and any attached files.

Reviewer #1: No

Reviewer #2: No

Reviewer #3: No
---

## [Editor Report · Decision Letter 2]

18 Jan 2021

Dear Dr Raugei,

We are pleased to inform you that your manuscript 'Mechanical Coupling in the Nitrogenase Complex' has been provisionally accepted for publication in PLOS Computational Biology.

Best regards,

Turkan Haliloglu

Associate Editor

PLOS Computational Biology

Nir Ben-Tal

Deputy Editor

PLOS Computational Biology

---

## [Editor Report · Acceptance letter]

27 Feb 2021

PCOMPBIOL-D-20-01220R2 

Mechanical Coupling in the Nitrogenase Complex

Dear Dr Raugei,

I am pleased to inform you that your manuscript has been formally accepted for publication in PLOS Computational Biology. Your manuscript is now with our production department and you will be notified of the publication date in due course.

With kind regards,

Alice Ellingham
